# Angle-programmed tendril-like trajectories enable a multifunctional gripper with ultra-delicacy, ultrastrength, and ultraprecision

Yaoye Hong [1], Yao Zhao[1], Joseph Berman[2], Yinding Chi[1], Yanbin Li[1], He (Helen) Huang [3,4] & Jie Yin [1] ✉

Achieving multicapability in a single soft gripper for handling ultrasoft, ultrathin, and ultraheavy objects is challenging due to the tradeoff between compliance, strength, and precision. Here, combining experiments, theory, and simulation, we report utilizing angle-programmed tendril-like grasping trajectories for an ultragentle yet ultrastrong and ultraprecise gripper. The single gripper can delicately grasp fragile liquids with minimal contact pressure (0.05 kPa), lift objects 16,000 times its own weight, and precisely grasp ultrathin, flexible objects like 4-μm-thick sheets and 2-μm-diameter microfibers on flat surfaces, all with a high success rate. Its scalable and material-independent design allows for biodegradable noninvasive grippers made from natural leaves. Explicitly controlled trajectories facilitate its integration with robotic arms and prostheses for challenging tasks, including picking grapes, opening zippers, folding clothes, and turning pages. This work showcases soft grippers excelling in extreme scenarios with potential applications in agriculture, food processing, prosthesis, biomedicine, minimally invasive surgeries, and deep-sea exploration.

Soft robotic grippers have emerged as a promising approach for manipulating various objects using fluid-driven soft actuators[1–6], underactuated compliant elastomers[7], origami[8]/kirigami structures[9,10], and stimuli-responsive materials[11–17]. These grippers leverage softness to adapt to different targets[5,18], facilitating pinching[9,17], enveloping[7], suction[5,8], and entangling[6] grasps. Their adaptivity enables grasping objects in a safe and delicate manner. For example, a delicate dielectric elastomer actuator (DEA)-based gripper can grasp fragile objects[19] such as raw eggs and deformable water-filled balloons via pinching. A hydraulic ribbon-based soft robotic gripper[3] can ultragently manipulate fragile gelatinous organisms such as live jellyfish via enveloping. Softness enhances delicacy, however, forfeits strength and precision[20–23], as exhibited by the low payload-to-weight ratios of 1 and 80 in the delicate hydraulic[3] and DEA-based[19] grippers, respectively. Such small forces hinder their ability to handle heavy objects.

To bridge the gap between softness and strength, researchers proposed different designs for achieving low-to-high grasping strength, including suction[5], fluid-driven rigidity percolation[1,18,24], and varied stiffness of stimuli-responsive materials[16,17]. For example, a five-finger-shaped gripper based on shape memory polymers enables manipulating heavy objects with a record-high payload-to-weight ratio up to 6400[17]. However, the gain in strength often sacrifices delicacy for nondestructive manipulation, e.g., suction and jamming-based soft grippers cannot grasp gelatinous organisms noninvasively[1,18,24]. Moreover, when objects are tiny or thin, precision grasp remains challenging given their small contact constraints[25]. To pick up small objects, extended appendages are needed, as demonstrated in the kirigami shell-based gripper capable of pinching a grain of sand[9]. For thin sheet objects, electroadhesion is used but with limitations that require smooth and dry flat surfaces[23].

[1]Department of Mechanical and Aerospace Engineering, North Carolina State University, Raleigh, NC 27695, USA. [2]Department of Electrical and Computer Engineering, North Carolina State University, Raleigh, NC 27695, USA. [3]UNC-NC State Joint Department of Biomedical Engineering, North Carolina State University, Raleigh, NC 27695, USA. [4]UNC-NC State Joint Department of Biomedical Engineering, University of North Carolina at Chapel Hill, Chapel Hill, NC 27599, USA. ✉e-mail: jyin8@ncsu.edu

The expanding frontier of robotic-grasping applications in biomedicine, deep-sea exploration, agriculture, food processing, minimally invasive surgeries, and prosthetics[20–23,26] requires grippers to specialize across high-requirement tasks, a substantial challenge that current grippers have not effectively addressed. For example, biofluidic manipulation requires ultradelicate grippers capable of handling extremely fragile objects such as droplets[27]. Minimally invasive surgeries require soft grippers to be high-delicate, strong, and high-precision[28]. The need for ideal robotic prosthetics requires soft robotic grippers equipped with analogous capabilities to human hands regarding dexterousness, precision, and load-carrying capacity[29]. Meanwhile, it requires soft grippers to be lightweight and easily integrated with prosthetics for simple and controllable actuation. However, bulky systems in pneumatic or hydraulic grippers[1,3,5,6] make it challenging.

Given the tradeoff between delicacy, strength, and precision, it remains challenging to simultaneously achieve high delicacy, high strength, and high precision in a single soft gripper[23,30] (see Supplementary Note 1 and Supplementary Table 1 for summary of representative soft grippers). One potential solution is the shape-morphing kirigami approach[9,10]. Our recent work in design of an encapsulating kirigami-inspired gripper takes a step toward a potential solution[10]. We demonstrated its proof-of-concept applications in manually picking a raw egg yolk nondestructively, a human hair, and heavy objects with a payload-to-weight ratio up to 1000[10]. It underscores the significance of morphologies in potentially mitigating the tradeoff challenges. However, its performance still falls short of achieving the goal of an extraordinary soft gripper that specializes across the versatile high-requirement tasks mentioned above (Supplementary Note 1 and Supplementary Table 2). Several challenges necessitate further exploration. These include effectively handling objects even more delicate than the raw egg yolk[10], such as extremely fragile water droplets. Furthermore, it poses significant hurdles, regarding addressing the ultraprecise manipulation of micro-size and thin objects thinner than a human hair, accommodating ultra-heavy objects that may surpass the reported record-high payload-to-weight ratio of 6400[17], as well as the potential integration of the gripper with prosthetic hands for multifunction.

Another potential solution is to mimic natural organisms, which balance this tradeoff by morphing with nastic trajectories to perform grasping and climbing, e.g., cucumber tendrils and cephalopod tentacles[31]. Soft grippers aim to imitate their shapes, but few explored leveraging the nastic trajectory for exceptional grasping capabilities[32]. Previous studies in soft grippers emphasize the adaptive morphologies for improved capabilities[23,30]. Their grasping trajectories receive less attention and remain largely unexplored[33]. However, the significance of the trajectory should not be overlooked, as it profoundly influences the gripper's performance, especially when dealing with noninvasive manipulation of fragile objects and precision grasping of small/thin objects[23]. Explicitly programming trajectories pose challenges due to the inherent complexities arising from the nonlinear deformations in compliant structures[34] (Supplementary Note 2).

In this work, we propose combining adaptive morphologies in kirigami and nastic curves in tendril plants to address the challenges. Here, we report an angle-based kirigami gripper by manipulating the angle $\gamma_o$ in the X-shaped ribbon (Fig. 1a) for specializing across versatile high-requirement tasks. It shows unprecedented capabilities compared to the state-of-the-art soft grippers[23,30] (refer to Supplementary Note 1 for an in-depth discussion). First, it enables programmable tendril-like trajectories by $\gamma_o$ in an explicit and controlled manner. Second, it can simultaneously achieve ultragentle yet ultrastrong and ultra-precision manipulations. We show that it is capable of noninvasively grasping extremely soft objects, e.g., a water droplet with nearly zero stiffness (Fig. 1b and Supplementary Movie 1), precisely grasping an ultra-thin polymer sheet as thin as 4 μm and 2 μm-diameter microfiber that are 20 times and 40 times thinner than a typical human hair, respectively, and strongly grasping an ultra-heavy dead weight 16,000 times the self-weight of the gripper (0.4 g), which is 2.5 times the reported record-high payload-to-weight ratio[17] (Fig. 1c and Supplementary Movie 1). Third, we demonstrate a proof-of-concept environment-friendly gripper made of biodegradable materials such as bare leaf (Fig. 1d) in ultragentle grasping of a dandelion (Fig. 1e) and other objects (Supplementary Movie 2). This green design philosophy can minimize both the impact on targets and the ecological footprint[35]. Fourth, the lightweight and simple displacement-controlled gripper facilitates its integration with prostheses without the need of additional tethered power and actuation systems in fluidic-driven grippers[1,3,5,6]. We demonstrate its integration with robotic prosthetics and multifunctionality in handling various challenging delicate tasks (Supplementary Movie 3), including noninvasively picking a grape from the vine (Fig. 1f), opening a zipper, and turning a book page etc. Feedback systems required for delicate tasks in existing prostheses[36] are not necessary due to the tendril-like trajectory.

## Results

### Programmable trajectories through manipulating $\gamma_o$

Figure 1a shows the 2D precursor design of the gripper. It comprises a thin flat sheet patterned with parallel cuts, where a central X-shaped ribbon forms an original intersecting angle, $\gamma_o$. It is fabricated through laser cutting of a thin polyethylene terephthalate (PET) sheet with a thickness of 127 μm (see "Methods" section). Uni-axial stretching changes the intersecting angle from $\gamma_o$ to $\gamma$ (inset of Fig. 2a). $\gamma$ increases towards 180° with the applied strain $\varepsilon$. Correspondingly, the gripper deploys in such a way that ribbons buckle into a pop-up shell-like caging shape composed of two shells as grasping petals bridged by two cones (Fig. 1b, c, e, f). The buckling ribbons resemble the shape of Euler elastica[37–39]. We combine analytical analysis, experiments, and finite element method to elucidate how the trajectory programmed by $\gamma_o$ paves the way towards noninvasive and delicate grasping.

For the angle-based design, the original ($\gamma_o$) and deformed ($\gamma$) intersecting angle can be explicitly related to the applied strain $\varepsilon$ (or the displacement). The relationship is expressed as

$$\varepsilon = c_r \left( \frac{\sin(\gamma/2)}{\sin(\gamma_o/2)} - 1 \right) \tag{1}$$

where $c_r$ is a constant related to the normalized length of the stretching ribbon (Supplementary Note 2). The angle design yields the explicit relationship, where a small variation in the curved boundary with the continuity conserved barely affects the performance. Such relationship facilitates the displacement control of the trajectory as discussed later. In contrast, despite the generation of various 3D curved shapes in the boundary curvature-based design in our previous study[10], its inexplicit and nonlinear relationship between the displacement and the Gaussian curvature makes it not suitable for the explicit displacement control (Supplementary Note 2).

Tracing the movement of the end effectors of the gripper in the $yz$-plane defines the grasping trajectory during the stretching-induced deployment. This trajectory plays an important role in governing the grasping performances.

By approximating the shape of the discrete ribbons in the gripper as Euler elastica[37–39], we can obtain the Cartesian coordinates along the trajectory curve expressed as

$$\bar{y}_t = \left( \frac{2}{\lambda} E\left( AM(\lambda \bar{s}_h, m_s), m_s \right) - \bar{s}_h \right) \cos(\beta) \tag{2}$$

$$\bar{z}_t = \left( \frac{2}{\lambda} E\left( AM(\lambda \bar{s}_h, m_s), m_s \right) - \bar{s}_h \right) \sin(\beta) \tag{3}$$

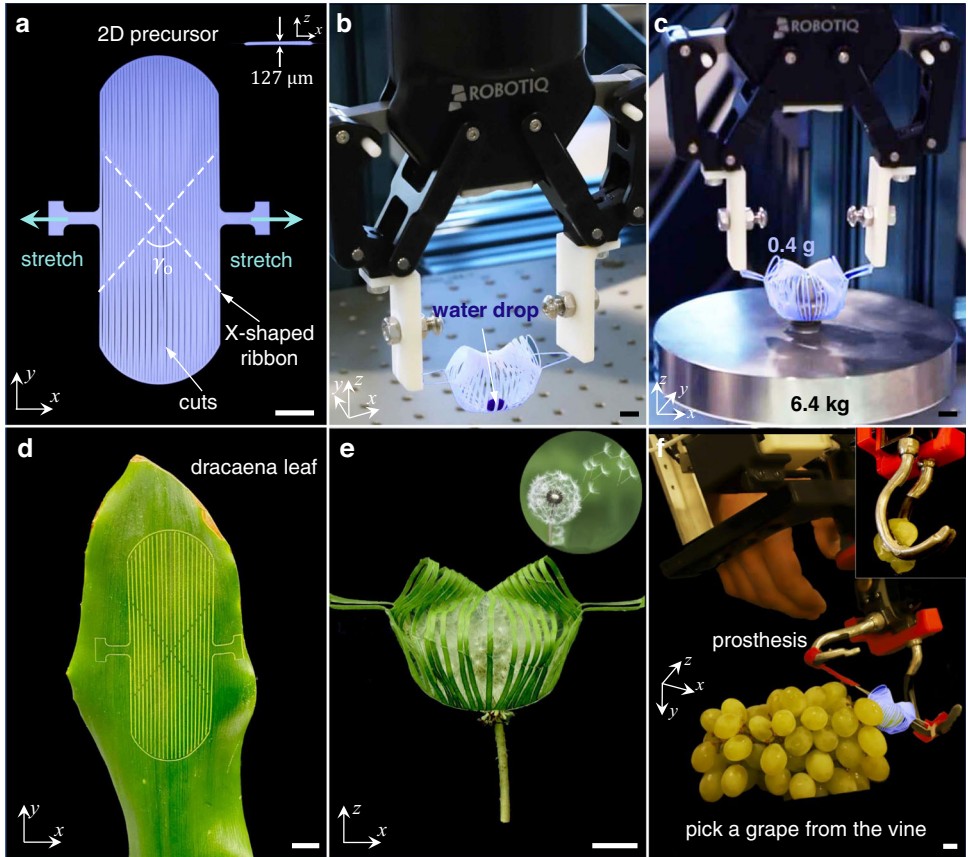

**Fig. 1 | Demonstration of a noninvasive and biodegradable kirigami gripper.**
**a** The 2D precursor of the kirigami gripper is divided by a central X-shape ribbon network into two fan shapes and two triangles patterned with parallel cuts. The angle $\gamma_o = 80°$ is highlighted in dashed white lines, with the inset showing the side view with a 127 μm thickness. Green arrows are the direction of uniaxial stretching. **b**, **c** The kirigami gripper integrated with a robotic arm grasping a water droplet (**b**)

and a 6.4 kg heavy deadweight (**c**), 16,000 times the weight of the gripper. **d**, **e** 2D precursors of the gripper made of a bare leaf (**d**). The gripper made of a leaf grasps a dandelion (**e**). **f** The kirigami gripper integrated onto a prosthesis electrical terminal device (ETD) assists the prosthesis with picking a grape from the vine noninvasively. The inset shows the squeezed ruptured grape after picking without using the kirigami gripper. Scale bars = 10 mm.

where $\bar{y}_t$ and $\bar{z}_t$ are the coordinate functions (Supplementary Note 2). $\bar{s}_h$ is half of the length of the longest discrete ribbon in the petal normalized by the half-width of the 2D precursor. $\lambda = 2F(\frac{\pi}{2}, m_s)/\bar{s}_h$. E, F, and AM denote the elliptic functions. $m_s(\gamma_o, \varepsilon)$ and $\beta(\gamma_o, \varepsilon)$ denote the elliptic modulus and rotating angle of the discrete ribbons in the petals, respectively, which depend on $\gamma_o$ and $\varepsilon$ (Supplementary Fig. 3).

Figure 2a shows the analytically predicted curved trajectories of various grippers with different $\gamma_o$. First, for grippers with a large $\gamma_o$ (e.g., $\gamma_o = 170°$), the limited variation in $\gamma$ from 170° to 180° leads to a short trajectory similar to a circular arc. The arc's $z$-axis displacement decreases monotonically with $\varepsilon$. The deployed gripper exhibits a 3D shape with two petals far from each other at the maximum applied strain $\varepsilon_{max}$. Second, a slight decrease in $\gamma_o$ (e.g., $\gamma_o = 150°$) extends the trajectory path to be longer (Fig. 2a). Compared to $\gamma_o = 170°$, the gap between the two petals at $\varepsilon_{max}$ is smaller. Figure 2b shows that the experimental side-view deploying process is consistent with the analytical prediction. At a relatively small applied strain (e.g., $\varepsilon \sim 0.11$), the grippers deform into a large-angled V-shape, when the two petals remain flat and the central parts pop up into cones. At an intermediate applied strain (e.g., $\varepsilon \sim 0.23$), the two petals gradually bend into a shell-like shape and rotate to approach each other. Third, for grippers with a further decreased $\gamma_o$ (e.g., 130°), the trajectory curls more and exhibits a critical ω-shape. The two petals initially bend and approach the lowest position ($y = -1$), and subsequently lift to meet in the middle to close the gap (Fig. 2a). For grippers with a small $\gamma_o < 90°$ (e.g., $\gamma_o = 10°$ and 80°), a larger variation in $\gamma$ with $\varepsilon$ causes ribbons to bend more. It makes the ω-shaped trajectory more prominent. Figure 2c shows the

experimental deploying process of the gripper with $\gamma_o = 80°$, which agrees well with the analytical model. It exhibits a similar deploying process to that of $\gamma_o = 130°$ but with a smooth spherical caging shape at $\varepsilon_{max}$.

We utilize $\gamma$ to bridge the gap between the curvature $\bar{\kappa}$ of the trajectory and the strain $\varepsilon$ (or displacement). Figure 2d shows that the variation in $\bar{\kappa}$ increases approximately linearly with $\varepsilon$,

$$\bar{\kappa} \propto \varepsilon \qquad (4)$$

in the trajectory curve of the gripper with $\gamma_o = 80°$ (Supplementary Note 3). It facilitates the explicit control of the trajectory through $\varepsilon$. Interestingly, all the trajectories in Fig. 2a resemble the shape of the Euler spiral[40], consistent with the curve of a curling tendril (Fig. 2d) in plants[41]. Mimicking the nastic morphology of plants, the gripper with a gradually curling trajectory enables an ultra-gentle touch, as discussed later. This biomimetic trajectory with gradually increasing curvature $\bar{\kappa}$ is first explored and utilized in soft grippers, to our knowledge.

To quantify how $\gamma_o$ programs the grasping trajectory, we derive the correlation between $\gamma_o$ and the normalized maximum curvature $\bar{\kappa}_{max}$ of the trajectory curve (Fig. 2a) at $\varepsilon = \varepsilon_{max}$. The correlation is expressed as

$$\bar{\kappa}_{max} \propto \sqrt{180° - \gamma_o} \qquad (5)$$

Figure 2e shows that $\bar{\kappa}_{max}$ is proportional to the square root of the variation in angle 180°-$\gamma_o$. The intersecting angle starts with $\gamma_o$ and

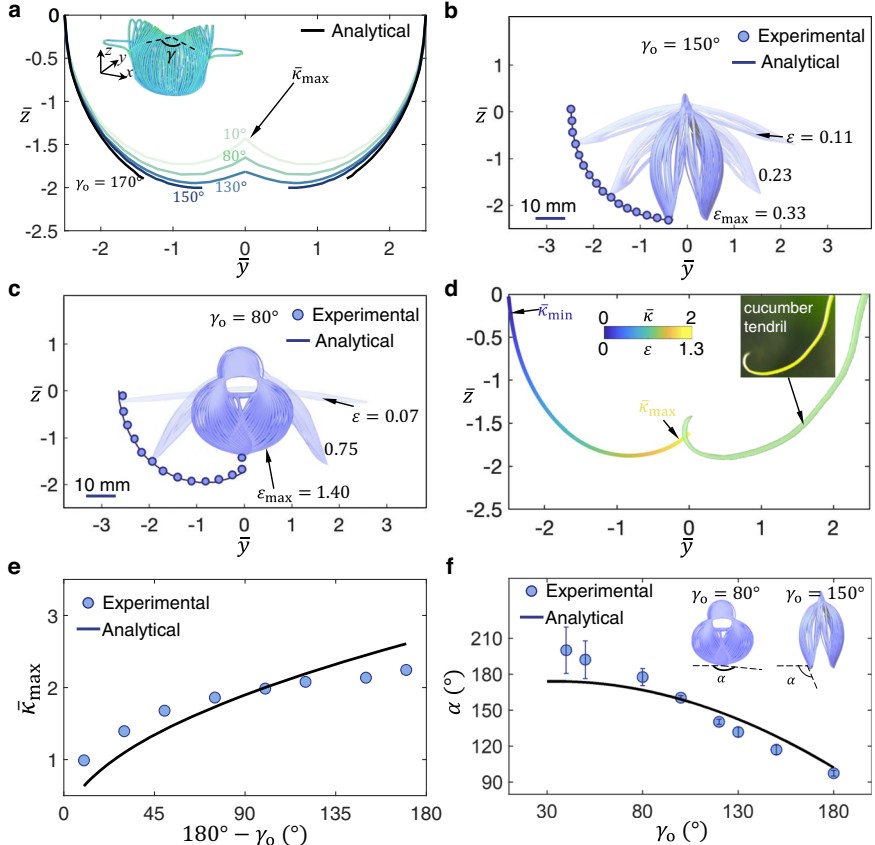

**Fig. 2 | Programmable and controllable grasping trajectories of the gripper.**
**a** The deploying trajectories at the end effector of petals in the grippers with different original intersecting angles $\gamma_o$. The color of the curves represents different $\gamma_o$. $\bar{\kappa}_{max}$ denotes the normalized curvature of the trajectory curve at the maximum applied strain $\varepsilon_{max}$. The inset shows the isometric view of the simulation results of the gripper with $\gamma_o = 80°$, with $\gamma$ denoting the changing intersecting angle upon stretching. **b**, **c** Side-view (in the $yz$-plane) closure process of the grippers with $\gamma_o = 150°$ and $80°$. The deploying trajectories at the end effector of petals with an increasing applied strain $\varepsilon$. **d** The trajectory of the gripper with mimicking the cucumber tendril curve. The color bar represents the curvature $\bar{\kappa}$ along the trajectory proportional to the applied strain $\varepsilon$. $\bar{\kappa}_{max}$ and $\bar{\kappa}_{min}$ are the maximum and minimum curvature along the trajectory. The inset shows a cucumber tendril. **e** The normalized curvature $\bar{\kappa}_{max}$ of the trajectory curve at the maximum applied strain as a function of the variation $(180°-\gamma_o)$ in the original intersecting angle. **f** The grasping/closing angle $\alpha$ as a function of the angle $\gamma_o$ in different 2D precursors. The grasping angle $\alpha$ denotes the angle between the tangential direction at the end tip of the petal and the horizontal axis in the fully deployed state. Schematics show the side-view of different $\alpha$ demonstrated by grippers with $\gamma_o = 80°$ and $150°$. The error bars are the standard errors of the mean. Scale bars = 10 mm.

ends at 180° during the deployment (Supplementary Note 2). Overall, the smaller the angle $\gamma_o$ is, the curlier the trajectory (Fig. 2a).

Overall, Eq. (5) enables the simple program of the trajectories through manipulating $\gamma_o$. It leads to unprecedented grasping capabilities discussed later. Equation (4) allows for the explicit control of the trajectory curvature through the displacement or strain $\varepsilon$. Consequently, based on Eqs. (4)-(5), the trajectories become programmable and controllable. Noteworthily, both programming and controlling the trajectory are difficult for existing kirigami grippers based on a shell structure[9] and a curvature-based design[10]. It is mainly due to the nonlinear deformation of shells and the implicit relationship between the boundary curvature and trajectory (Supplementary Note 2). These significantly impede the gripper's performance in delicate tasks, particularly when integrated with robotic arms and prostheses that rely on a cost-effective displacement/position control[20–22]. Additionally, this contrasts the prevalent soft grippers based on pneumatic/hydraulic-driven[1–6] and stimuli-responsive materials[11–17], where explicitly programming their grasping trajectories is very challenging due to the high material nonlinearity.

**Programmable grasping angles through manipulating $\gamma_o$**

In addition to the trajectories, the grasping angle $\alpha$ between the two end effectors determined by the fully deployed petals, plays a key role in delicate grasping. $\alpha$ is defined as the angle between the tangential direction at the end tip of the petal and the horizontal axis (insets of Fig. 2f). Theoretically, $\alpha$ can be predicted as

$$\alpha = -2\tan^{-1}\left(-\frac{2\sqrt{m_s^2 - m_s^4}}{1 - 2m_s^2}\right) - \bar{l}_r \tan^{-1}\left(-\frac{2\sqrt{m_c^2 - m_c^4}}{1 - 2m_c^2}\right) \quad (6)$$

$$\frac{2E(\frac{\pi}{2}, m_s)}{F(\frac{\pi}{2}, m_s)} - 1 = \frac{(1 + c_b)\sin(\frac{\gamma_o}{2}) - c_b}{\sin(\frac{\gamma_o}{2}) + \cos(\frac{\gamma_o}{2})} \quad (7)$$

where $m_s$ and $m_c$ denotes the elliptic modulus of the discrete ribbons in the spherical petals and central cones, respectively, which are correlated by $\gamma_o$ in Eq. (7). $\bar{l}_r$ is the ratio between the characteristic length of the ribbons of the spherical and conical regions (Supplementary Note 3). $c_b$ is a constant related to the buckling of the boundary ribbon.

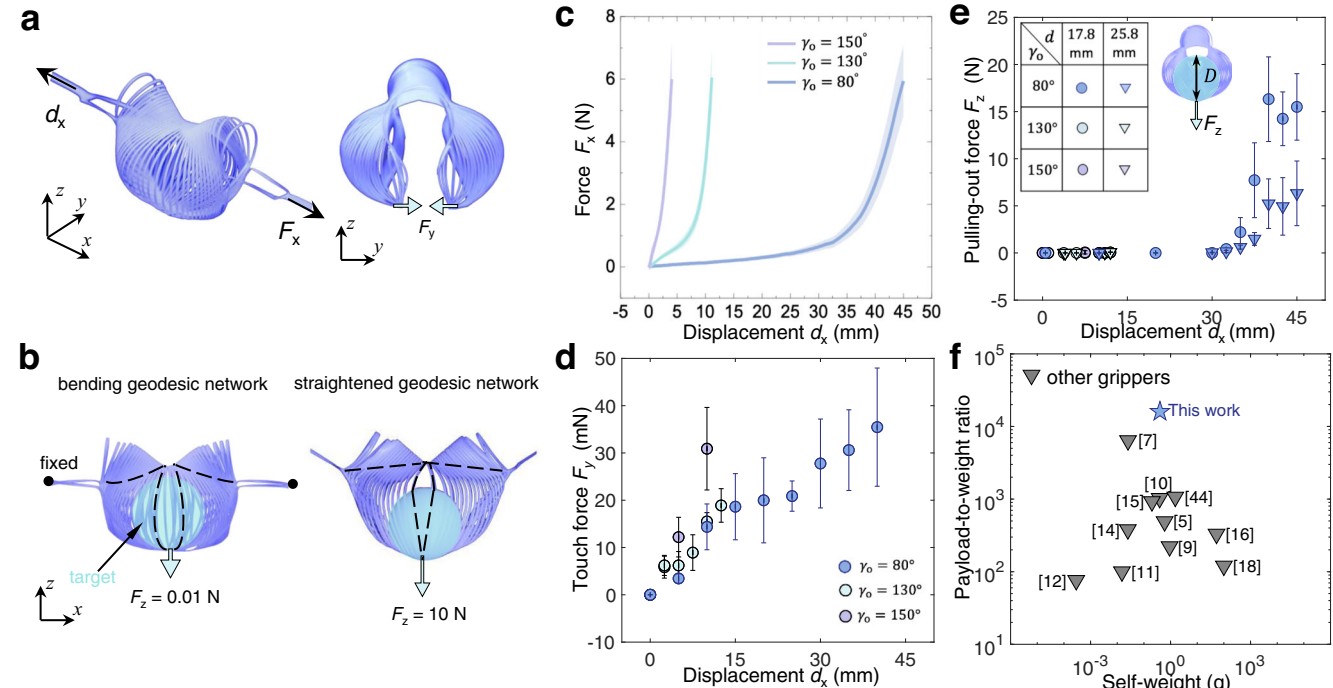

**Fig. 3 | Janus-faced grasping feature. a** The isometric view and the side view of the deploying gripper with the increasing displacement along the $x$-axis. $F_x$ denotes the applied stretching force along the $x$-axis. $F_y$ denotes the $y$-direction reaction force between the petals and target object when the gripper is in contact with the target. **b** Schematics showing the transition of the geodesic ribbon network (highlighted in dashed black line) from bending to stretching with an increasing pulling-out force $F_z$. $F_z$ denotes the force required in the $z$ direction to pull out the green sphere from the gripper, with the green sphere being the target object. Green arrows are the direction of the pulling-out force $F_z$. Black dots are the fixed points. **c** Experimental

force-displacement ($F_x$-$d_x$) curves for three grippers with $\gamma_o = 80°$, 130°, and 150°. The error bars represent the standard errors of the mean. **d** Experimental touch force $F_y$ as a function of the displacement $d_x$ of three grippers. **e** Experimental pulling-out force $F_z$ as a function of the displacement $d_x$ for the grippers. The inset shows the schematic illustration of measuring $F_z$ via pulling green spheres with two different diameters $D$ out of the grippers. The error bars represent the standard errors of the mean. **f** Comparison of the robustness (payload-to-weight ratio vs. self-weight) between our gripper and other works.

Figure 2f shows that as $\gamma_o$ decreases from 180° to 30°, $\alpha$ is predicted to monotonically increase from 97° to a plateau value of 180°, which is consistent with the experiment. However, due to the contact in the petals, experimentally, $\alpha$ can be even beyond 180° for a gripper with $\gamma_o$ close to 30°. Especially, the gripper with $\gamma_o = 80°$ show a grasping angle of close to 180° (i.e., $\alpha \sim 180°$). It defines a critical state, where the two petals form a smooth sphere-like shape (Fig. 2c). This is distinct from the sharp-angle tips formed by the two shallow-shell ($\gamma_o = 130°$, $\alpha = 131.9°$ and $\gamma_o = 150°$, $\alpha = 117.1°$) petals (Fig. 2b and Supplementary Fig. 4). The deployment with $\varepsilon$ in the three grippers can be well reproduced by the corresponding FEA simulations (Supplementary Fig. 4). It shows that despite the large applied strain, the maximum principal strain remains small (below ~1%) in the gripper due to the significant strain release through buckling of ribbons.

**Implications for nondestructive tasks**

With $\gamma_o$ programming the deploying trajectory and the grasping angle, the grasping capability evolves. For the grippers with a larger $\gamma_o$, their near circular-arc trajectories and non-smooth enclosing space formed by sharp-angle tips (i.e., a small $\alpha$) make the invasion unavoidable when pinching the target objects. By contrast, for the grippers with $\gamma_o = 80°$, the gradually curling trajectory and a near-180° grasping angle facilitate encapsulating a target object ultra-gently. Mimicking a cucumber tendril's nastic and gentle motion, the gradual curling trajectory minimizes the horizontal interaction between the petals and the object, as discussed next. This is especially suitable for noninvasively grasping extremely soft and fragile objects such as liquid droplets (Fig. 1b) and gelatinous organisms, discussed later. Additionally, for the grippers with $\gamma_o < 80°$, the curlier trajectory and the grasping angle

>180° disturb the smoothness of the spherical cage formed by petals (Supplementary Note 2). It undermines the grasping performance. Additionally, when the petals are compressed by the surface, $\alpha$ further increases to be closer to 180° before the closure of the petals. The deformation due to the compression makes the petals to approach the target in a more parallel way. Overall, considering the trajectory, the grasping angle, and the enclosing volume of petals (Supplementary Fig. 4), $\gamma_o = 80°$ is the optimal angle for a gripper to specialize across different extreme scenarios.

**Implications for ultragentle yet ultrastrong grasping**

To better understand the Janus-faced feature of being ultragentle to handle extremely soft objects (Fig. 1b) and ultrastrong to handle ultraheavy objects (Fig. 1c), we further explore the effects of $\gamma_o$ on the stretching force $F_x$ in the $x$ direction (Fig. 3a) and the two reaction forces $F_y$ (Fig. 3a) and $F_z$ (Fig. 3b) acting on the petals in the $y$ and $z$ direction during grasping. As $\gamma_o$ decreases from 150° to 80°, they show similar J-shaped $F_x$-$d_x$ curves but with a dramatically reducing initial stiffness (Fig. 3c). We note that reducing $\gamma_o$ has the similar effect on the $F_x$-$d_x$ curves as reducing the boundary curvature from positive to negative in our previous boundary curvature-based shape-morphing kirigami sheets[10]. For $\gamma_o = 80°$, a small stretching force $F_x$ of 2 N can result in a large displacement in the gripper. Such a small force facilitates its integration with robotic arms and prosthesis.

Before the closure of spherical petals, the inexpensive bending with the gradual curling trajectory leads to an ultragentle touch. Figure 3d shows the touch force $F_y$ (i.e., the quasi-static reaction force in the $y$ direction between the petal and target objects shown in Fig. 3a) increases with the $x$-axis displacement $d_x$ regardless of $\gamma_o$ but remains

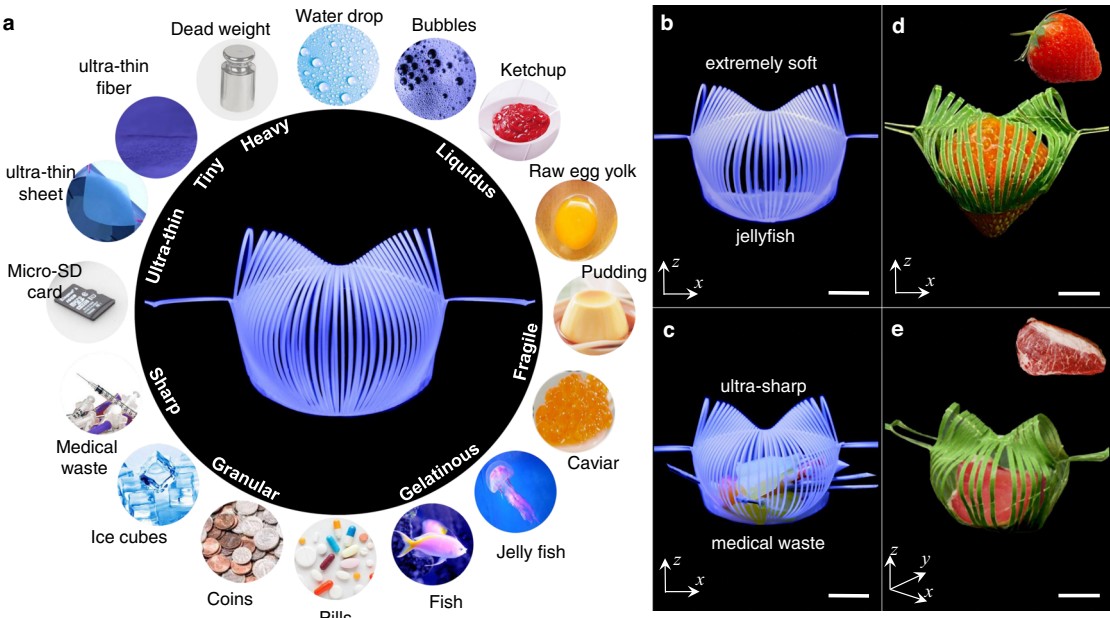

**Fig. 4 | Universality of the gripper. a** A collection of various objects in different forms, shapes, stiffness, sizes, and weight that the gripper can handle shows its universality. **b**, **c** Demonstration of the gripper in grasping a jellyfish and sharp granular medical trash. **d**, **e** The gripper made of a leaf grasps a strawberry and a piece of meat. Scale bars = 10 mm.

small with a maximum value of 35.5 mN. Notably, the contact pressure exerted by the spherical petals is about 0.0468 kPa, which is comparable to the reported gentlest gripper (0.0455 kPa) in handling a jelly fish[3].

After the closure of the petals, we explore its adaptive capability for low-to-high payload by pulling out an encapsulated sphere from the fully deployed gripper (Fig. 3b). The pulling-out force $F_z$ is defined as the minimum force required to pull the sphere out of the gripper. The pulling-out test simulates the scenario when a grasped heavy object tries to slip out of the closed petals. When the pulling force or the grasped object is relatively small or lightweight, the bending geodesic[42,43] network remains curved (Fig. 3b). However, when $F_z$ is large (Fig. 3b), the angle-based X-shape coupled with the curled-up trajectory at $\varepsilon_{max}$ makes the ribbon network become straightened with dominated stretching energy to prohibit the target from escaping. This bending-to-stretching energy evolution enables the gripper to hold the target object firmly, leading to a large pulling-out force. Figure 3e shows $F_z$ vs. displacement $d_x$ for pulling two different-sized spheres out of the grippers with different $\gamma_o$. For the gripper with $\gamma_o = 80°$, as $d_x$ increases from 32.5 mm to 40 mm, $\alpha$ approaches 180°, and the energy evolution in the spherical petals results in a jump of $F_z$ from an average force of 0.41 N to 16.3 N, a 39 times force enhancement. Note that the sphere with a large diameter ($D = 25.8$ mm) leads to a smaller pulling-out force than that with a smaller diameter ($D = 17.8$ mm) because the large diameter prohibits the closure of the sphere formed by the petals. In contrast, the maximum $F_z$ for the grippers with $\gamma_o = 130°$ and $\gamma_o = 150°$ is < 0.3 N (Fig. 3e), over 50 times smaller than that of the gripper with $\gamma_o = 80°$. This is because their uncurled trajectory ($\gamma_o = 130°$ and 150°) makes it energetically easier for the target to bend the petals for escaping.

We note that our previous work on the curvature-based design[10] prohibits deployed grippers from forming straightened geodesic ribbons. In contrast, the angle-based design (X-shape) and the curled-up trajectory yield a straightened geodesic network. It results in a bending-to-stretching energy evolution for unprecedented payload capacity shown in Fig. 1c. Thus, for the gripper with $\gamma_o = 80°$, the bending-to-stretching energy evolution enables it to be ultragentle in

the $y$ direction and ultrastrong in the $z$-direction, where $F_z$ is over 460 times larger than $F_y$ in the fully deployed state.

Next, we further compare the payload-to-weight ratio[21] of our gripper ($\gamma_o = 80°$) with that of a host of soft grippers[5,9–12,14–18,44] by categorizing them in a diagram of payload-to-weight ratio versus the mass of the gripper in Fig. 3f. The payload-to-weight ratio (i.e., the ratio of the maximum weight the gripper can carry and the mass of the gripper) is of paramount importance, especially when grippers are integrated with prostheses and autonomous vehicles[21]. Existing soft grippers harness fluid-driven rigidity percolation[1,24], variation in the stiffness of responsive materials[16,17], eletroadhesion[44], and suction[5] to overcome the intrinsic compliance in soft grippers[45,46]. We note that the payload-to-weight ratio of these reported grippers is entangled below 7000. Our kirigami gripper can achieve a record-high payload-to-weight ratio of 16,000, which is 2.5 times higher than the reported highest ratio of 6400[17] (yielded by a gold layer), and 16 times higher than that of our previous curvature-based work[10].

## Universality of the gripper
Leveraging the nastic trajectory, we further demonstrate the universality of the kirigami gripper in grasping a variety of objects in different forms, shapes, sizes, stiffness, materials, and weight, as summarized in Fig. 4a. First, different from pinching-based[9] grasping in most of the state-of-art grippers such as granular jamming[1], the unique curling-up trajectory similar to gentle tendrils of plants minimizes pinching and pressurizing the grasped objects. This is beneficial for noninvasively manipulating extremely soft and fragile objects, including liquids with close-to-zero stiffness, such as a water droplet, liquid bubbles, and non-Newtonian liquids with a hydrophobic coating, e.g., ketchup (Supplementary Fig. 5) and raw egg yolk, soft gels such as pudding and caviar (Supplementary Fig. 5), live gelatinous organisms such as jellyfish (Fig. 4b and Supplementary Movie 4) and fish. Second, the net-like enclosed structure can also well handle granular objects in different shapes, such as pills, coins, and ice cubes (Supplementary Fig. 5), as well as medical waste, such as needles and sharps, where gaps between the flexible ribbons can accommodate the randomly oriented sharps without damaging the gripper (Fig. 4c). This

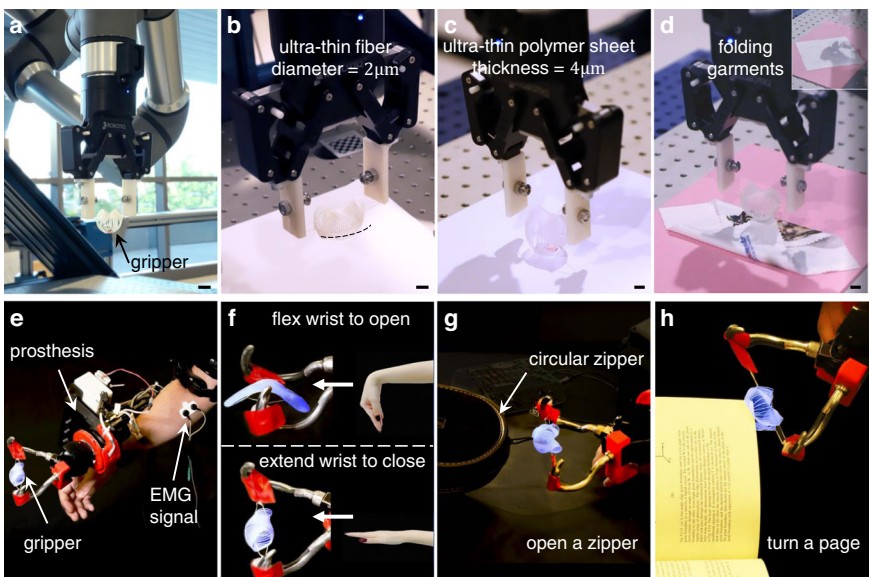

**Fig. 5 | Kirigami gripper integrated with a robotic arm and an electromyographic prosthesis. a–d** The kirigami gripper integrated onto a commercially available robotic arm grasps an ultra-thin fiber (**b**), an ultra-thin polymer sheet (**c**). The gripper folds garments with the inset showing the unfolded garments (**d**). **e–h** The kirigami gripper integrated onto a prosthesis electrical terminal device (ETD) with closed and open states (**f**) controlled by electromyographic (EMG) signals of the wrist flexor and extensor, respectively. The kirigami gripper assists the prosthesis with opening a zipper (**g**) and turning a book page (**h**). Scale bars = 10 mm.

is challenging for pneumatic/hydraulic-based soft grippers with the potential of sharps-induced leaking damages.

Third, the gradually curling-up trajectory and close-to-180° closing angle of the two end effectors make it easy and beneficial to grasp thin disks (e.g., micro-SD card) and delicately handle ultra-thin fibers and sheets, as discussed later. This is extremely challenging for granular jamming-based grippers and pneumatic grippers. Fourth, the bending-to-stretching deformation transition in the ribbons of grasping petals can largely enhance the holding force to pick up heavy objects (e.g., 6.4 kg dead weight in Fig. 1c). Besides mimicking the nastic curve of plants, the gripper could be made from plants or wood (e.g., the dracaena leaf in Fig. 1d) and could grasp fragile organisms, including dandelion (Fig. 1e), fragile fruits (e.g., strawberry in Fig. 4d), and slippery meat (Fig. 4e) for food processing to minimize the ecological footprint[35] of the soft robot during both manufacturing and grasping.

### Integration with robotic arms and prosthetic hands
Leveraging the programmable and controllable trajectories, lastly, we demonstrate the gripper's unaltered capability when it is integrated with both a commercial robotic arm (Fig. 5a–d) and an electromyographic (EMG)-controlled upper limb prosthesis (see "Methods" section) for complementing their functions in accomplishing challenging manipulation tasks (Fig. 5e–h).

Distinct from the reported soft grippers integrated onto a robotic platform[1,3,9], the tendril-like trajectory enables the gripper to perform universal feats that no current rigid or soft machine can accomplish. It can handle the uncertainty in the nondestructive and delicate grasping profoundly. For example, the same robotic kirigami gripper can successfully pick up a water droplet from a hydrophobic surface and place it on either hydrophobic or hydrophilic surfaces (Fig. 1b and Supplementary Movie 1), an ultra-thin fiber, e.g., a fiber thread with a diameter of 2 μm in Fig. 5b, which is 40 times thinner than a human hair (a diameter of around 80 μm), an ultra-thin sheet, e.g., a 4-μm thick polymer sheet in Fig. 5c, as well as heavyweight in Fig. 1c. The nastic trajectory enables the spherical petals to scoop the sheet from the target surface ultra-gently (see Supplementary Note 4 for detailed

discussions). The curling-up grasping, similar to the gentle tendrils, is demonstrated in Supplementary Fig. 5. It minimizes the horizontal interaction between the petals and the thin sheet. Its nastic trajectory and the spherical petals equip the gripper with the capability of handling multidimensional uncertainty in a passive and adaptive way. Thus, it makes the need of highly accurate control of the robotic arm unnecessary, which is especially beneficial when visual perception is hard to be achieved during manipulation[21,47]. Moreover, Fig. 5d shows that it can also manipulate thin and flexible objects, such as folding a soft square-shaped garment into a smaller folded square sheet, which are challenging tasks for most existing robotic manipulators[21,47]. The ultra-soft touch prevents damage to the target object without sacrificing the success rate of grasping, 95.2%.

The state-of-art prosthetic devices rely on the user's integration of sensory systems and motor control to achieve nondestructive functions, especially in dexterous tasks[36,48]. We first integrate a soft gripper onto prostheses, which provides an alternative solution to ensure noninvasive interaction in universal tasks. Though existing soft grippers gain non-destructivity but forfeit the universality[3,7,17], here, the programmable and controllable trajectory enables our kirigami gripper to assist the prosthesis users in universal and nondestructive tasks. Crucially, the simple and explicit relationship, Eqs. (4)-(5), stemming from the X-shape design, eases the trajectory control when it is integrated with displacement-controlled prostheses. The easily-deployed feature facilitates the integration with any prosthetic hands (including our demonstrated electrical terminal device (ETD)). When integrated with an EMG-controlled prosthesis ETD (Fig. 5e and Supplementary Movie 3), the kirigami gripper switches between the open and close states with the ETD's movements operated by the EMG signals[49–51] recorded from the wrist extensor and flexor, respectively. Flexing and extending the wrist (Fig. 5f) correspond to the open and closed state of the kirigami gripper, respectively (see "Methods" section).

Further, we evaluate the significantly improved performance of the prosthesis with a kirigami gripper in several representative delicate tasks in daily life, such as picking fruits, opening a zipper, and turning a book page. Figure 1f shows that the kirigami gripper can successfully pick a grape from the vine easily and nondestructively (Supplementary

Movie 3), where prostheses without a high-quality sensory system may squeeze and rupture the grape (inset of Fig. 1f). The spherical petals of the kirigami gripper help it encapsulate the grape with a high success rate of 78.6%. Figure 5g shows that the kirigami gripper can also successfully open a circular zipper (Supplementary Movie 3). For existing prostheses without feedback control, slipping out is unavoidable due to the changeable reaction force when manipulating a straight zipper, let alone a circular zipper. Here, the curled-up trajectory coupled with the spherical petals enables the gripper to fit the uncertain reaction passively. Furthermore, Fig. 5h shows that the prosthesis with a kirigami gripper can easily turn a book page smoothly. Similar to grasping a super-thin sheet, the gradually curling trajectory enables the gripper to pick and turn a page, which is challenging for the existing robotic prosthesis hands and other soft grippers with unprogrammed trajectories using pinching[9] or enveloping[7]. Moreover, the bulky support systems of pneumatically or hydraulically driven[1,3,5,6] grippers make it challenging to be integrated onto a wearable prosthesis.

## Discussion

Predominant soft grippers neglect the function of trajectories, due to the complexity of explicitly driving artificial trajectories towards nastic morphologies in nature. It results in systems incapable of specializing across high-requirement tasks. In this article, we proposed a simple angle-based design strategy for a noninvasive, ultrastrong, and universal gripper with high precision. Combining experiments and theoretical modeling, we showed that the shapes of tendril-like trajectories can be explicitly programmed by the angle $\gamma_o$ and simply controlled by the applied strain $\varepsilon$ or displacements. Emulating the natural curve enables the gripper to achieve preternatural grasping performances, i.e., extremely soft and ultrathin\tiny targets. The programmable and controllable tendril-like trajectories can be well predicted by the developed analytical modeling, which is validated by related experiments. Further, the programmed trajectory yields a bending-to-stretching energy evolution during heavy-target grasping, thereby radically augmenting its grasping capability. The powerful gripper is capable of grasping various extreme objects in terms of stiffness, size, geometry, form, and weight, including an ultrafragile water drop and gelatinous jellyfish, an ultrathin 4μm-thick polymer sheet and 2μm-diameter fiber, an ultraheavy dead weight that is over ten thousand times its self-weight, and sharp granular medical trash, etc.

We anticipate that the gripper could have broad applications in robotics, marine organism protections, agriculture, food processing, prosthesis, and medical devices. The design philosophy that treats trajectories as mutable leads to functional (i.e., grasping) and energetic evolution. It will catalyze the next-generation soft machine systems. Further, the proof-of-concept demonstration as a biodegradable gripper using bare leaf represents an opportunity for rethinking the engineering design of soft machines that can better integrate with our ecosystem.

In comparison to state-of-the-art specialized and universal soft robotic grippers[23], this work fills the important knowledge gaps both fundamentally and practically. Fundamentally, this work overcomes the challenges in theoretically predicting the grasping trajectories of the reported kirigami grippers[9,10], as well as other soft grippers made of fluidic elastomers[1–6] or stimuli-responsive materials[11–17]. The intricate nature of trajectories in these soft grippers stems from the nonlinear and large deformations inherent in compliant structures and soft materials. Practically, for the first time, it unifies ultradelicacy, ultrastrength, ultraprecision, universality, and multifunctionality in one single gripper, which is not achieved in all the reported soft robotic grippers (see detailed comparison and discussions in Supplementary Note 1, Supplementary Tables 1, 2). When it comes to delicate tasks such as water droplets and jellyfish, achieving noninvasive grasping presents a significant challenge due to the uncertainties inherent in dynamic task environments and the unpredictable reactions of living organisms (Supplementary Note 4). However, by programming the trajectory, we enhance our gripper's ability to handle the uncertainties associated with uncertain environments.

This gripper is scale- and material-independent. A question remains regarding what material could optimize durability and strength. Materials with larger Young's modulus and fracture toughness would improve the performance but requires a higher energy input during stretching. Moreover, to minimize the disturbance of the smoothness in the curved surface, the upper and lower boundary requires a $C^2$ continuity (Supplementary Note 4). For a boundary with a $C^1$ or $C^0$ continuity, varying the geometry of localized ribbons at the discontinuous point in the boundary could improve performance for extremely soft objects. The aspect ratio and the width of the cut in the 2D precursor are optimized to improve the grasping capability (Supplementary Note 4). The grasping performance is affected by the relative size of the target, a large size could make the gripper unable to encapsulate the target. It could cause the petals to pinch the target, accompanied by a sharp drop in the pulling-out force from ~ 15 N to ~ 0.5 N (Supplementary Note 4). Additionally, the maximum principal strain $\varepsilon_{max}$ in the popping-up ribbons is small, resulting in over 1,000 repeated cycles of 1 kg deadweight lifting with the gripper without failure. Round tips at the cut tips can be applied to relieve the stress concentration for improving durability.

## Methods

### Fabrication and mechanical testing of the kirigami sheets
We used polyethylene terephthalate (PET) sheets from Dupont Teijin Film (McMaster–Carr) to make the kirigami sheets. The Young's modulus, Poisson's ratio, and thickness are 3.5 GPa, 0.38, and 0.127 mm, respectively. The 2D precursors of the grippers with various cut patterns were cut utilizing a laser cutter, EPILOG LASER 40 Watts. The ribbon width of 1.5 mm in Fig. 1a. The force-displacement curves are derived based on uniaxial tensile tests using Instron 5944 with a loading rate of 10 mm per min. The touch force and pulling-out force are also tested using Instron 5944.

### Demonstration of the kirigami gripper
For the grasping demonstrations, the grippers were uni-axially stretched either manually or by a commercially available robotic arm (Robotiq 2F-85 gripper integrated onto the UR5e robotic arm from Universal Robotics). When integrated with the robotic arm, the kirigami gripper requires a 3D-printed device to be connected with the Robotiq 2F-85 gripper for grasping heavy objects. The 3D printed device is made of VeroWhite and 3D printed by Stratsys, Objet 260. The target ultra-thin polymer sheet (Lumirror film #4 - F56 thickness 4 μm) used for demonstration is produced by Toray Industries, Inc. The ultra-thin fiber (diameter 2 μm) is an individual filament from the Polyester 300D fiber.

### Integration with prosthesis device
When integrated with the prosthesis device, the kirigami gripper is controlled by EMG signals. Surface EMG signals were collected from gelled, bipolar electrodes placed over the extensor carpi radialis longus (ECRL) and flexor carpi radialis (FCR) muscles, identified via palpation. The electrodes were connected to an EMG system (MA400, Motion Lab Systems, Inc., USA), and the EMG signals were recorded at a 1000 Hz sampling frequency. The magnitude of each EMG signal proportionally drove the speed of the prosthesis ETD motor in one direction.

### Finite element simulation
In the finite element method (FEM) simulation (Abaqus/Standard), the kirigami sheets of the grippers were modeled as isotropic, linear elastic material (Young's modulus of 3.5 GPa, Poisson's ratio of 0.38). We used solid quadratic tetrahedral elements (C3D10H) to mesh the shapes.

Also, the fine mesh was applied to the connecting regions between the discrete ribbons. The right end was fixed, and a prescribed displacement was applied to the left end.

## Data availability

The authors declare that the data supporting the findings of this study are available within the article and its Supplemental Information files. Extra data are available from the author upon request.

## Code availability

The code used for the analyses will be made available upon e-mail request to the corresponding author.

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

## Acknowledgements
The authors acknowledge the help from Drs. Y. Lee, P. Cohen, and Mr. T. Batchelder on the integration with the robotiq arm. The authors acknowledge the funding support from the National Science Foundation under award number CMMI-2005374 (J.Y.) and IIS-2221479 (J.Y. and H.H.).

## Author contributions
Y.H. and J.Y. designed research. Y.H. conducted the theoretical modeling and robotic demonstrations. J.B. and H.H. designed and conducted the prosthesis demonstration. Y.H. and Y.Z. performed finite element simulation. Y.Z., Y.C., and Y.L. conducted the mechanical testing. All authors analyzed the data. Y.H. and J.Y. wrote the paper and all the co-authors revised the paper.

## Competing interests
The authors declare no competing interests.
