## [Peer Review File · Nature Communications]

Reviewers' comments:

Reviewer #1 (Remarks to the Author):

Hong et al. demonstrate a nice approach to making tendril-like grippers and using them for robotic applications. The paper reported angle-programmed tendril-like trajectories that enable an ultragentle yet ultrastrong and delicate gripper, leveraging morphogenesis intelligence for programming grasping morphology and trajectory to achieve a versatile soft gripper that is ultrastrong and delicate.

However, the author in previous work has been published in Nature communications (doi.org/10.1038/s41467-022-28187-x) entitled "Boundary curvature guided programmable shape-morphing kirigami sheets", ([29] as cited), where the deformation of this gripper has been described in a fundamental mechanism by the Gauss-Bonnet theorem, and in that article also included analytical modeling, numerical simulation, and experiments, the reviewer suggests to further elaborate the novelty of the presented work.

The authors explored a simple angle-based design strategy for a kirigami gripper with high delicacy. However, it is difficult for us to find the differentiation and strength of this manuscript compared to previously reported paper. For example, in gripper performance, it is necessary to clearly communicate what differentiation (actuation response time, weight ratio, etc.) is compared to the various previously reported soft grippers.

The gripper in the citation mentioned on page 13 does not need to programming the gripping trajectories and can pick up fragile objects only through compliance and adaptivity, and adjusting the number and size of grippers in citation 16 can also achieve this function, please be more specific about the new contribution of the proposed gripper.

Overall, the similarity of the previous work and novelty, we recommend transfer another specific aim`s journal.

Reviewer #2 (Remarks to the Author):

This paper introduces a Kirigami-based soft gripper, which can handle various objects ranging from a delicate object to a very heavy object compared to the weight of the gripper. The gripper utilizes the optimized trajectory of the gripper tip, minimized lateral force for gripping for a delicate object, yet strong lifting force due to the leveraged tensile strength of the material itself. The gripper shows a significant gripping performance in terms of delicacy and strength. I left minor comments to the authors for further improvement of the paper.

Q1. The authors show the key design parameter as γ_0 , which is the angle of the uncut lines of the 2D precursor. But I guess that there would be other design parameters which can affect the gripping behaviors. For example, upper and lower boundary shape of the precursor (the curve on the 2D precursor), the thickness of the cuts, and the aspect ratio of the 2D precursor. Could the authors give some details on these design parameters about how they will affect the gripping?

Q2. The gripper seems to be bigger than the object to grip fully. But there would be some cases that the gripper cannot fully engulf the object. In that case, the gripper mouth will be open, but the tip is holding the object. Could the authors elaborate what will be the performance of the gripper when it is not fully closed, but gripping an object using the shear force at the tip?

Q3. When gripping an object, especially a very thin object (as demonstrated like ultra-thin fiber), aligning the gripper tip exactly on the surface of the substrate of the object would be challenging. I guess a practical strategy would be pushing the gripper towards the substrate which can ensure the contact to the substrate, yet it deforms the gripper due to the compression. In such a case, how will the gripping performance change in terms of gripping a very thin object?

Reviewer #3 (Remarks to the Author):

I have read the manuscript "Angle-programmed tendril-like trajectories enable an ultragentle yet ultrastrong and delicate gripper" with great interest.

As suggested by the title, the work focuses on the design of a soft gripper capable of handling delicate objects. While the manuscript is rather clear and beautifully illustrated I don't believe that the work is suitable for Nature Communications. In effect, similar designs have been published already, most notably by Holmes and his group in Science Robotics (2021).

While I appreciate that the authors cite this work in their manuscript, and I am sure could argue that their approach surpasses that of Holmes and collaborators in some metrics; I simply see the reported work as incremental, thereby contradicting the novelty criterion required for publishing in this journal.

Response to all reviewers

In response to the comments from the reviewers, we have revised the title and the first seven paragraphs of the manuscript, and added a new section entitled “Supplementary note 1: Comparison with existing grippers”, two new supplementary tables, and five new supplementary figures from Fig. S6 to Fig. S10 in the supplementary materials.

Previous title: Angle-programmed tendril-like trajectories enable an ultragentle yet ultrastrong and delicate gripper

New title: Angle-programmed tendril-like trajectories enable a **multifunctional gripper with ultradelicacy, ultrastrength, and ultraprecision**

The main finding of our paper is that we overcome the tradeoff between delicacy, strength, and precision in the current state-of-the-art soft robotic grippers. This breakthrough is achieved through the utilization of angle-programmed tendril-like trajectories. It enables us to unify ultradelicate, ultrastrong, and ultraprecision grasping within a single gripper. The revised introduction makes our finding explicit.

Before we get to the individual comments of the reviewers, we summarize the novelty and significance of our work.

Novelty. We demonstrate for the first time a single, ultragentle yet ultrastrong and ultraprecision universal gripper. It is capable of simultaneously grasping extremely soft water droplets with nearly zero stiffness without damage, an ultraheavy object that is 16,000 times its self-weight (2.5 times the reported record-high payload-to-weight ratio), and challenging micro-size objects such as an ultra-thin flexible polymer sheet with a thickness of 4 μm and a 2 μm -diameter microfiber, both of which are 20 times and 40 times thinner than a typical human hair (80- μm -diameter). No existing research has reported universal soft robotic grippers that achieve a comparable level of delicacy, strength, and precision. No prior studies have documented specialized soft grippers that exhibit comparable levels of either ultradelicacy or ultrastrength or ultraprecision. This is particularly true when successfully picking up a waterdrop or an extremely thin 4 μm flexible sheet from flat surfaces with a consistently high success rate. Further, we make a significant contribution by identifying, for the first time, the critical role played by adaptive morphology-coupled tendril-like trajectories in enabling these unprecedented capabilities. The angle-programmed trajectories can be explicitly predicted through analytical modeling. It addresses the challenge in trajectory prediction in the widely used soft grippers made of soft elastomers and compliant structures due to their highly nonlinear and large deformation. Moreover, we demonstrate for the first time an eco-friendly delicate soft gripper made of biodegradable natural leaves by utilizing the material- and size-independent design. Lastly, we demonstrate for the first time easy integration with robotic prosthetic hands in assisting in handling challenging tasks in daily life, such as picking a grape from a vine, turning a page, opening a zipper, and folding clothes, etc. No work in robotic prosthetic hands have reported such comparable capabilities.

Significance. The expanding frontier of robotic-grasping applications specializes across biomedicine, deep-sea biological exploration and conservation, agriculture, food processing, prosthetics, and minimally invasive surgeries. The applications require soft robotic grippers to specialize across versatile high-requirement tasks. These tasks encompass a wide range of specific requirements, such as ultradelicacy in handling extremely fragile droplets in biomedicine and organisms in marine, as well as the simultaneous need for delicacy and strength in agricultural applications. Additionally, biocompatibility, delicacy, and strength are essential in food processing, particularly when dealing with slippery meat and similar materials. Furthermore, integrating lightweight grippers with prosthetic hands is crucial for handling dexterous tasks in daily life. Lastly, biomedical devices necessitate grippers that exhibit ultraprecision, delicacy, and strength. Despite recent advances in the field, effectively balancing strength, delicacy, and precision remains a substantial hurdle for current specialized or universal grippers. This challenge primarily stems from the tradeoff between strength, delicacy, and precision, due to the compliance in soft grippers. Our

work overcomes this tradeoff and unifies them in one single gripper with significantly enhanced delicacy, strength, precision, versatility, and multifunctionality in handling versatile extreme scenarios. It greatly expands the frontier of soft robotic grippers in specializing across the aforementioned versatile high-requirement tasks. In addition, the design and mechanisms of programmable tendril-like trajectories in this work can apply to grippers with micro or macro-scale, e.g., noninvasive manipulation of fragile micro-organisms, etc., and different materials, e.g., natural eco-friendly materials for reducing the ecological footprint.

We next summarize the main issues raised by the reviewers and our responses.

Responses to the reviewers’ concerns on novelty. The reviewers mentioned two papers (2021 Sci. Robot. and our 2022 Nat. Commun. paper) that report kirigami-based grippers. On close comparison with these, we respectfully disagree that these papers challenge the novelty and significance of our work.

Before we comment on these papers one by one, we first introduce the motivation of our work by highlighting the grand challenge in the broader context of soft robotic grippers, including not only the kirigami grippers but also other reported grippers based on different designs and actuation mechanisms.

Soft robotic grippers have significantly expanded the horizons of conventional rigid robots due to their materials and structural compliance. This advancement has led to the development of a diverse library of specialized or universal soft grippers for delicate or versatile tasks. These grippers find extensive applications in various domains, including agriculture, food processing, deep-sea biological exploration, minimally invasive surgeries, prosthetics, and more. Compliance, while facilitating morphology adaptivity, necessitates a tradeoff that results in compromises in terms of strength and precision. Numerous studies have been dedicated to overcoming this tradeoff, yet they have thus far yielded solutions that prioritize either strength, or delicacy and strength, or precision, or precision and delicacy. **To date, unifying delicacy, strength, and precision within one single soft gripper has yet to be realized** due to the tradeoff. It largely hinders their applications in specializing across versatile applications and presents a grand challenge for soft robotic grippers.

Next, we comment on these paper one by one under such a motivation.

1) **Yi Yang, Katherine Vella, and Douglas P. Homes, Grasping with kirigami shells. Science Robotics, 6 (54), eabd6426 (2021).**

Fig. R1 Comparison between 2021 Sci. Robot. and this work.

First, we compare the robotic performances with both Yang et al.'s work and other reported state-of-the-art soft grippers.

Based on the classical periodic cut pattern in kirigami, Yang et al.'s paper reported a simple kirigami gripper with the demonstrated delicacy of grasping a strawberry and precision of grasping a grain of sand and a flat disk-shaped pill via a pinching mode (Fig. R1, left), which are challenging for traditional soft grippers. However, **they need three different kirigami grippers with modified designs to achieve either delicacy or precision but not both in one single gripper (Fig. R1, left).**

In contrast, based on a new and different X-shaped cut pattern (Fig. R1, right), our work show that a single gripper with the same design is capable of ultragently grasping extremely fragile objects such as a water droplet with nearly zero stiffness, precisely picking up a 4 μm -thin flat polymer sheet (over 250 times thinner than a typical pill with a thickness of ~ 1 mm) and a 2 μm -diameter microfiber, and strongly picking up and holding an ultra-heavy object (6.4 kg or 14.1 lb) that is 16,000 times its self-weight. Different from the pinching mode in Yang et al.'s gripper, our gripper employs an encapsulating or enveloping mode that enables extreme delicacy and precision. **For the first time, we unify not only ultra-delicacy but also ultra-precision and ultra-strength in one single gripper.**

Compared to Yang et al.'s work and other reported state-of-the-art soft grippers, our gripper exhibits superior individual performance in terms of delicacy, precision, or strength. For delicacy, previous studies report a dielectric elastomer actuator (DEA)-based soft gripper capable of grasping a fragile, highly deformable water-filled thin membrane balloon using a pinching mode (Shintake et al., Adv. Mater. 2016, 28, 231), as well as a hydraulic ribbon-based soft robotic gripper capable of ultragently manipulating fragile gelatinous organisms such as a live jellyfish using enveloping grasping (Sinatra et al., Sci. Robot. 4, eaax5425, 2019). The contact pressure is extremely small, 0.0455 kPa in Sinatra et al.'s gripper for ultragentle manipulation. In this work, we choose the extremely fragile case of a water droplet without a thin membrane and also a jellyfish using an encapsulating mode. Our gripper can also generate an extremely small contact pressure less than 0.05 kPa for ultragentle grasping. It is comparable to the reported gentlest gripper (0.0455 kPa) in handling a jellyfish by Sinatra et al.

For precision, Yang et al.'s work needs a long extended appendage to pick up a sand grain. The sizes of our grasped thin sheet and micro-fiber are over 20 and 40 times thinner or smaller than a typical human hair, respectively.

For strength, Roh et al. reported a five-finger-shaped soft gripper based on shape memory polymers capable of manipulating a heavy object with a record-high payload-to-weight ratio up to 6400 (Roh et al., Sci. Robot. 6, eabi6774, 2021). Our gripper is 2.5 times the record-high payload-to-weight ratio.

Second, the fundamental science and mechanisms governing the two different pinching and encapsulating modes in Yang et al. and our work are distinct. For a pinch mode, its grasping trajectory plays a negligible role compared to its adaptive morphology. In contrast, for an encapsulating mode, its grasping trajectory of the end effectors plays a critical role in the ultra-delicate and ultra-decision grasping of water droplets, thin sheets, micro-fibers, etc., regarding how it approaches and grasps the objects, which is not studied in Yang et al.'s work.

The modeling of grasping trajectories remains largely unexplored due to the challenge in the highly nonlinear materials properties and large deformation in soft grippers. We show that in this work the tendril-like grasping trajectory underpins the observed unprecedented performances in ultraprecision and ultradelicacy. The angle-programmed tendril-like trajectories can be well predicted through developed analytical modeling validated by experiments.

Third, we further expand the important multifunctionality of our grippers in eco-friendly delicate grippers to reduce the ecological footprint as demonstrated in natural leaf- and wood-based delicate grippers, as well as easy integration with industrial robotic arms and prosthetic hands in handling challenging and delicate

tasks in daily life, such as picking a grape from a vine, turning a page, opening a zipper, and folding clothes, etc. These key demonstrations highlight a step toward practical applications of our gripper, which are reported for the first time.

In summary, based on the above reasons and comparison with the state-of-the-art soft grippers, we think our work represents a breakthrough and a significant advance in soft grippers in terms of unifying ultradelicacy, ultrastrength, ultraprecision, versatility, and multifunctionality in one single gripper, as well as the fundamental understanding of the underpinned theoretically predictable tendril-like grasping trajectories. Thus, we respectfully disagree that our work is “similar to” Yang et al.’s work and “incremental”, as Reviewer 3 commented.

2) **Yaoye Hong, Yinding Chi, Shuang Wu, Yanbin Li, Yong Zhu, Jie Yin, Boundary curvature guided programmable shape-morphing kirigami sheets, Nature Commun. 13, 530 (2022)**

Fig. R2 Selected results on boundary curvature-guided shape-morphing kirigami sheets in our 2022 Nat. Commun. paper

Our last year’s work focused on the boundary-curvature-guided shape-shifting in kirigami from 2D precursors with different boundary curvatures to 3D curved surfaces with different Gaussian curvatures (Fig. R2). Kirigami grippers and dynamically conformable heaters for human knees serve as two examples of proof-of-concept applications by leveraging its adaptive morphology. Our last year’s work in boundary curvature-guided design of an encapsulating kirigami gripper takes a step toward a potential delicate, strong, and precision gripper. We demonstrated its proof-of-concept applications by manually picking up a fragile raw egg yolk nondestructively, an 80- μm -diameter human hair without the need for additional attachments, and a heavy object with a payload-to-weight ratio up to 1000. It suggests the vital role of adaptive morphologies in potentially balancing the tradeoff.

Despite the advance, several knowledge gaps remain to be filled to address the grand challenge.

First, regarding the performance, it is still far from realizing the target high-performance soft gripper that specializes across versatile high-requirement tasks. Several challenges persist in the field, necessitating

further exploration. These challenges include noninvasively handling extremely fragile objects such as water droplets, which are softer than the raw egg yolk and other reported water-filled balloon (Shintake et al., Adv. Mater. 2016, 28, 231) without a thin membrane. Additionally, achieving ultraprecision is essential to manipulate micro-size and thin objects, surpassing the thickness of human hair. Moreover, it requires ultrastrength to accommodate ultra-heavy objects that may surpass the reported record-high payload-to-weight ratio of 6400 (Roh et al., Sci. Robot. 6, eabi6774, 2021). Lastly, the potential multifunctionality of integrating the gripper with prosthetic hands for handling diverse and complex tasks adds to the hurdles. **To conquer these challenges, a novel and comprehensive solution is required regarding both designs and fundamental mechanisms.**

Second, the current state-of-the-art researches on soft grippers focus on leveraging the grasping morphology for adaptivity, whereas they pay little attention to the grasping trajectory. As mentioned above, grasping trajectory plays a key role in delicate and precision grasping, especially for enveloping grasping. It is not studied in our previous study. However, theoretical modeling of grasping trajectories is very challenging in soft grippers. This is due to the complexities in explicitly modeling and programming the trajectories. These complexities arise from the highly nonlinear and large deformation in the soft materials and compliant structures. Specially, the implicit relationship between the boundary curvature and trajectory of our reported curvature-based kirigami gripper makes it rather challenging to program and control the grasping trajectories.

In the current work, the new angle-based kirigami gripper shows superior performances to our previous grippers, including using robotic arms rather than manual stretching in our previous work to pick and place droplets that are softer than egg yolk, 16 times higher payload-to-weight ratio, and a 4 μm -thin flat polymer sheet (20 times thinner than a human hair) and a 2 μm -diameter microfiber (40 times thinner than a human hair). The new angle-based kirigami gripper also enables programmable tendril-like trajectories by the angle in an explicit and controlled manner, theoretically and experimentally. We also expanded its multifunctionality in eco-friendly delicate grippers, and easy integration with prosthetic hands in handling challenging and delicate tasks in daily life.

We next address the comments of reviewers point by point.

Referee #1

Summary: “*Hong et al. demonstrate a nice approach to making tendril-like grippers and using them for robotic applications. The paper reported angle-programmed tendril-like trajectories that enable an ultragentle yet ultrastrong and delicate gripper, leveraging morphogenesis intelligence for programming grasping morphology and trajectory to achieve a versatile soft gripper that is ultrastrong and delicate.*”

Response: We thank the reviewer for the summary of our work. In the revised version, we have highlighted all the revisions in RED color.

Comment 1: “*However, the author in previous work has been published in Nature communications (doi.org/10.1038/s41467-022-28187-x) entitled “Boundary curvature guided programmable shape-morphing kirigami sheets”, ([29] as cited), where the deformation of this gripper has been described in a fundamental mechanism by the Gauss-Bonnet theorem, and in that article also included analytical modeling, numerical simulation, and experiments, the reviewer suggests to further elaborate the novelty of the presented work.*”

Response: We thank the reviewer for bringing up the comparison with our previous work for clarification.

The novelty of our work is highlighted from both analytical and application sides as below:

1) From the analytical perspective

In our previous work, we combined modeling, simulation, and experiment to study the boundary-curvature-guided shape-shifting from 2D kirigami precursors with positive, negative, and zero boundary curvatures to 3D curved surfaces with different Gaussian curvatures (see Fig. R2). However, we **did not fundamentally study the deformation of the gripper through any analytical modeling or numerical simulation** but rather the proof-of-concept demonstrations of the grippers. The deformations in the circular disk-shaped kirigami precursor and the gripper significantly differ from the gripper exhibiting more complex deformation.

The complexity arises from the gripper's intricate structure, specifically the interplay of multiple polygons formed by curved ribbons, which are missing in the circular disk-shaped kirigami precursor with only parallel ribbons. While the previous study has examined the deformation of individual ribbons, the presence of coupled polygons introduces additional challenges. Further, the programming and control of the grasping trajectory are complicated by the integral of the changeable curvature along the boundary, as dictated by the Gauss-Bonnet theorem.

In this work, the new angle-based design allows for simple programming of the grasping trajectory through an angle α_0 and explicit control of the trajectory through the applied strain ϵ . To our knowledge, this work represents the first successful implementation of explicit programming and control of grasping trajectories in soft grippers. The experimental results obtained align closely with both analytical and numerical analyses. Programming and controlling the trajectory leads to significantly improved grasping capability, as discussed later.

2) From the application perspective

In our previous work, we demonstrated a proof-of-concept kirigami gripper based on the curvature-based design, which can nondestructively grasp an egg yolk and a live fish. However, the unprogrammed trajectory inhibits the gripper's potential to specialize across different extreme scenarios in biology and biomedicine.

In this study, we programmably drive artificial trajectories towards nastic curve in nature. It leads to significantly improved grasping capability. Additionally, the controllable trajectory results in

unprecedented performances when integrated with robotic arms and prostheses, an aspect that was not explored in our previous work.

First, for noninvasive tasks, the new gripper's tendril-like trajectory surpasses its predecessor by enabling manipulation of water drops with nearly zero stiffness. This represents a significant advancement compared to the previous gripper, which was limited to handling gelatinous organisms.

Second, for heavy objects, the curled-up trajectory enables it to handle weights up to 16,000 times its own weight. This is 16 times higher than the previous one. Additionally, the bending-to-stretching energy evolution yielded by the curling-up trajectory enables a Janus-faced feature: it is ultrastrong while retaining the ultragentle feature. This bending-to-stretching energy evolution is first harnessed in soft grippers and generates a record-high payload-to-weight ratio.

Third, for ultra-thin targets, the gradually curling trajectory enables it to grasp a 2- μm -diameter fiber, 40 times smaller than the previous version (80- μm -diameter), as well as 4- μm -thick polymer sheet, not demonstrated before.

Fourth, when the gripper is integrated with the robotic arms and the prosthesis that rely on a cost-effective displacement control, the new angle-based design eases the displacement control of the tendril-like trajectory. This makes feedback control unnecessary in delicate tasks. For example, in noninvasively picking a grape from the vine, the success rate is nearly double that of the previous version. The integration with prostheses and robotic arms is never demonstrated in our previous work.

Overall, our new design surpasses the previous one by excelling in specialized performance across various extreme scenarios. These scenarios encompass the handling of objects that are extremely soft, ultra-heavy, and ultra-thin/tiny. A detailed comparison of the presented work and our previous work is provided in Table R1.

	Previous work	This work
Programmable deformation	✓	✓
Programmable and controllable trajectory	×	✓
Noninvasively manipulate gelatinous organisms	✓	✓
Noninvasively manipulate liquid objects	×	✓
Gentleness (contact pressure)	Unexplored	0.0468 kPa
Strength (maximum payload-to-weight ratio)	1,000	16,000
Delicacy (minimum resolution for ultra-thin objects)	80 μm	2 μm
Manipulate gelatinous organisms when integrated onto robotic arms and prostheses	×	✓
Manipulate liquid objects when integrated onto robotic arms and prostheses	×	✓

Table R1. A comparison between our previous work and this work

In response to the reviewer’s comment, in the revised version, we have added the following in the main text:

On Page 3 and 4, we added

“Inspired by the kirigami approach⁹, our recent work in boundary curvature-guided design of an encapsulating kirigami gripper takes a step toward a potential delicate, strong, and precision gripper¹⁰. We demonstrated its proof-of-concept applications in manually picking up a fragile raw egg yolk nondestructively, a 100- μm -diamater human hair without the need of additional attachments, and a heavy object with a payload-to-weight ratio up to 1000¹⁰. It underscores the significance of adaptive morphologies in potentially mitigating the tradeoff challenges. However, despite these advancements, its performance still falls short of achieving the ultimate goal of an extraordinary high-performance soft gripper that specializes across the diverse range of high-requirement tasks mentioned above. Several challenges persist in the field, necessitating further exploration. These include effectively handling objects that are even more delicate than the reported water-filled balloon¹⁹ and raw egg yolk¹⁰, such as water droplets. Additionally, addressing the manipulation of micro-size and thin objects thinner than a human hair that requires ultraprecision, and accommodating ultra-heavy objects that may surpass the reported record-high payload-to-weight ratio of 6400¹⁷, as well as the potential integration of the gripper with prosthetic hands for multifunctional capabilities, pose significant hurdles.”

“For example, the implicit relationship between the boundary curvature and trajectory of the curvature-based kirigami gripper in our previous study¹⁰ (Supplementary note 2) makes it rather challenging to program and control the grasping trajectories.”

“We show that it is capable of noninvasively grasping extremely soft objects, e.g., a water droplet with nearly zero stiffness (Fig. 1B and Supplementary Video S1), precisely grasping an ultra-thin polymer sheet as thin as 4 μm and 2 μm -diameter microfiber that are 20 times and 40 times thinner than a typical human hair, respectively, and strongly grasping an ultra-heavy dead weight with a 16,000 times the self-weight of the gripper (0.4 g) that is 16 times the load capacity in our curvature-based design¹⁰ and 2.5 times the reported record-high payload-to-weight ratio¹⁷ (Fig. 1C and Supplementary Video S1).”

On Page 16, we added

“Fundamentally, this work overcomes the challenges in theoretically predicting the grasping trajectories of the reported kirigami grippers^{9,10}, as well as other soft grippers made of fluidic elastomers¹⁻⁶ or stimuli-responsive materials¹¹⁻¹⁷ due to their high materials nonlinearity and large deformation.”

In the SI, we also added the above Table R1 and discussions on Page 4-6 in “Supplementary note 1. Comparison with existing grippers.”

Comment 2: “The authors explored a simple angle-based design strategy for a kirigami gripper with high delicacy. However, it is difficult for us to find the differentiation and strength of this manuscript compared to previously reported paper. For example, in gripper performance, it is necessary to clearly communicate what differentiation (actuation response time, weight ratio, etc.) is compared to the various previously reported soft grippers.”

Response: This work has distinct advantages over previously reported soft grippers, particularly in its ability to specialize across high-demand scenarios encountered in biology and biomedicine.

The comparison is provided in Table R2 and is discussed below.

	[1]	[3]	[7]	[14]	[15]	[16]	[17]	[9]	[10]	[5]	[44]	[46]	This work
Grasping mechanism	Jamming	Fluid-driven actuator	Fluid-driven actuator	responsive material	responsive material	responsive material	responsive material	Stretching-induced deformation	Stretching-induced deformation	Suction	Electro-adhesion	Adhesive material	Stretching-induced deformation
Programmable deformation	×	×	✓	×	✓	×	×	×	✓	×	×	×	✓
Programmable and controllable trajectory	×	×	×	×	×	×	×	×	×	×	×	×	✓
Noninvasively manipulate gelatinous organisms	×	✓	✓	×	×	×	×	×	✓	×	×	×	✓
Noninvasively manipulate liquid objects	×	×	×	×	×	×	×	×	×	×	×	×	✓
Gentleness (contact pressure)	Unexplored	0.0455 kPa	Unexplored	Unexplored	Unexplored	Unexplored	Unexplored	Unexplored	Unexplored	-15.7 kPa	Unexplored	2.5 kPa	0.0468 kPa
Strength (maximum payload-to-weight ratio)	Unexplored	Unexplored	<100	380	925	330	6,400	222	1,000	490	1,067	Unexplored	16,000
Delicacy (minimum resolution for ultra-thin objects)	Unexplored	Unexplored	Unexplored	Unexplored	Unexplored	10 μm	3.2 mm	1.5 mm	80 μm	2.5 mm	Unexplored	Unexplored	2 μm
Manipulate gelatinous organisms when integrated onto robotic arms and prostheses	×	✓	✓	×	×	×	×	×	✓	×	×	×	✓
Manipulate liquid objects when integrated onto robotic arms and prostheses	×	×	×	×	×	×	×	×	×	×	×	×	✓

Table R2. A comparison between other soft grippers and this work

Reference in Table R2 (citation number same as that in the main text):

- [1] Brown, E. *et al.* Universal robotic gripper based on the jamming of granular material. *Proceedings of the National Academy of Sciences* **107**, 18809-18814, doi:10.1073/pnas.1003250107 (2010).
- [3] Sinatra, N. *et al.* Ultragentle manipulation of delicate structures using a soft robotic gripper. *Science Robotics* **4**, eaax5425 (2019).

- [5] Song, S., Drotlef, D.-M., Son, D., Koivikko, A. & Sitti, M. Adaptive Self-Sealing Suction Based Soft Robotic Gripper. *Advanced Science* **8**, 2100641 (2021).
- [7] Teoh, z. e. *et al.* Rotary-actuated folding polyhedrons for midwater investigation of delicate marine organisms. *Science Robotics* **3**, eaat5276 (2018).
- [9] Yang, Y., Vella, K. & Holmes, D. Grasping with kirigami shells. *Science Robotics* **6**, eabd6426, doi:10.1126/scirobotics.abd6426 (2021).
- [10] Hong, Y. *et al.* Boundary curvature guided programmable shape-morphing kirigami sheets. *Nature Communications* **13**, 530, doi:10.1038/s41467-022-28187-x (2022).
- [14] Ma, M., Guo, L., Anderson, D. G. & Langer, R. Bio-inspired polymer composite actuator and generator driven by water gradients. *Science* **339**, 186-189, doi:10.1126/science.1230262 (2013).
- [15] Hubbard, A. M., Mailen, R. W., Zikry, M. A., Dickey, M. D. & Genzer, J. Controllable curvature from planar polymer sheets in response to light. *Soft Matter* **13**, 2299-2308, doi:10.1039/C7SM00088J (2017).
- [16] Linghu, C. *et al.* Universal SMP gripper with massive and selective capabilities for multiscaled, arbitrarily shaped objects. *Sci Adv* **6**, eaay5120 (2020).
- [17] Roh, Y. *et al.* Vital signal sensing and manipulation of a microscale organ with a multifunctional soft gripper. *Sci Robot* **6**, eabi6774, doi:10.1126/scirobotics.abi6774 (2021).
- [44] Cacucciolo, V., Shintake, J. & Shea, H. Delicate yet strong: Characterizing the electro-adhesion lifting force with a soft gripper. *2019 2nd IEEE International Conference on Soft Robotics* (2019).
- [46] Ruotolo, W., Brouwer, D. & Cutkosky, M. R. From grasping to manipulation with gecko-inspired adhesives on a multifinger gripper. *Science Robotics* **6**, eabi9773, doi:10.1126/scirobotics.abi9773 (2021).

1) Comparison with noninvasive grippers

We start by comparing our gripper with several notable soft and noninvasive grippers. For example, the kirigami gripper based on a rhombic shell (Yang *et al.*, *Science Robotics* 2021, Ref. 9 in the main text) can pick a raspberry using a pinching mode¹. The nondestructive gripper based on an origami polyhedron (Teoh *et al.*, *Science Robotics* 2018, Ref. 7 in the main text)² and hydraulic ribbons (Sinatra *et al.*, *Science Robotics* 2019, Ref. 3 in the main text)³ can grasp a jellyfish using an enclosing mode. Existing noninvasive grippers have been applied to fragile objects¹ and gelatinous organisms^{2,3}.

When considering the **noninvasive feature**, it becomes evident that existing grippers fall short in meeting the requirements of high-demand scenarios in biology, such as the nondestructive grasping of marine life that is softer than jellyfish, and in biomedicine, such as the noninvasive manipulation of liquid drops. This comparison is presented in detail in Table R2. Distinctly, our gripper distinguishes itself by excelling in the noninvasive aspect. It can nondestructively manipulate a liquid drop with near-zero stiffness, as shown in Fig. 1. This feature arises from programming the grasping trajectory toward the nastic curve in plants. The gradually curling-up trajectory enables an ultra-gentle touch.

Considering the **strong feature**, our work stands out with a remarkable payload-to-weight ratio of 16,000, surpassing other works as demonstrated in Table R2. This payload-to-weight ratio holds significant relevance for noninvasive tasks conducted in unstructured environments, as highlighted in the paper by Robert J. Wood *et al.* (*Science Robotics* 2022)⁴. In comparison, Yang *et al.*'s work achieves a payload-to-

weight ratio of approximately 222, attributed to the pinching grasping model they employ¹. The ratio in Teoh et al.'s work is less than 100 due to the rotating polyhedron made of rigid thermoplastic².

When considering the **design simplicity** aspect, it is worth noting that Teoh et al.'s work necessitates the use of supporting stents to enable the rotation of the origami polyhedron. Both Teoh et al.'s and Sinatra et al.'s work require the support of bulky external hydraulic systems^{2,3}. In Yang et al.'s work, three different design modifications are required for the delicacy of grasping a strawberry and the precision of grasping a grain of sand and a flat disk-shaped pill. They need three different kirigami grippers with modified designs to achieve either delicacy or precision but not both in one single gripper¹. In contrast, our work achieves specialization across different tasks with a single design. This distinct feature proves advantageous in assisting prostheses in delicate tasks, as demonstrated in Figure 1 and Figure 5. Existing prostheses often necessitate frequent changes to the end effector for different tasks. Our gripper's versatility eliminates this need.

2) Comparison with strong grippers

Compared with the reported strongest gripper using a gold layer coupled with shape memory polymers (Roh et al., *Science Robotics*, 2021, Ref. 17 in the main text)⁵, with a payload-to-weight ratio of 6,400, our gripper boasts a ratio that is 2.5 times higher, reaching an impressive 16,000. Compared to all existing strong grippers based on stimuli-responsive materials, suction, fluid-driven rigidity percolation, jamming, adhesive materials and electro-adhesion shown by Fig. 3F and Table R2, our gripper stands out with the largest payload-to-weight ratio.

3) Comparison with universal grippers

Existing universal grippers rely on various mechanisms, such as suction (Song et al., *Advanced Materials* 2021, Ref. 5 in the main text)⁶, responsive shape memory polymers (Linghu et al., *Science Advances* 2020, Ref. 16 in the main text)⁷, and jamming (Brown et al., *PNAS* 2010, Ref. 1 in the main text)⁸. However, none of them can achieve universal grasping while retaining the noninvasive feature. Specifically, suction-based grippers face limitations when it comes to preserving the integrity of gelatinous organisms. Shape memory polymers require a high temperature (e.g., 80°C in Linghu et al.'s work)⁷ that could harm living organisms, and negatively impact response times. The jamming structure falls short in its ability to grasp gelatinous organisms and liquid objects.

Regarding the actuation time, since our gripper is displacement-controlled, the actuation time is fast by simply stretching the gripper in the robotic arm, which is much faster than the stimuli-responsive soft materials based grippers.

The cited references in the response are listed as below:

Reference

- 1 Yang, Y., Vella, K. & Holmes, D. Grasping with kirigami shells. *Science Robotics* **6**, eabd6426, doi:10.1126/scirobotics.abd6426 (2021).
- 2 Teoh, z. e. *et al.* Rotary-actuated folding polyhedrons for midwater investigation of delicate marine organisms. *Science Robotics* **3**, eaat5276, doi:10.1126/scirobotics.aat5276 (2018).
- 3 Sinatra, N. *et al.* Ultragentle manipulation of delicate structures using a soft robotic gripper. *Science Robotics* **4**, eaax5425, doi:10.1126/scirobotics.aax5425 (2019).
- 4 Gruber, D. F. & Wood, R. J. Advances and future outlooks in soft robotics for minimally invasive marine biology. *Sci Robot* **7**, eabm6807, doi:10.1126/scirobotics.abm6807 (2022).

- 5 Roh, Y. *et al.* Vital signal sensing and manipulation of a microscale organ with a multifunctional soft gripper. *Sci Robot* **6**, eabi6774, doi:10.1126/scirobotics.abi6774 (2021).
- 6 Song, S., Drotlef, D.-M., Son, D., Koivikko, A. & Sitti, M. Adaptive Self-Sealing Suction-Based Soft Robotic Gripper. *Advanced Science* **8**, 2100641, doi:<https://doi.org/10.1002/adv.202100641> (2021).
- 7 Linghu, C. *et al.* Universal SMP gripper with massive and selective capabilities for multiscaled, arbitrarily shaped objects. *Sci Adv* **6**, eaay5120, doi:10.1126/sciadv.aay5120 (2020).
- 8 Brown, E. *et al.* Universal robotic gripper based on the jamming of granular material. *Proceedings of the National Academy of Sciences* **107**, 18809-18814, doi:10.1073/pnas.1003250107 (2010).

In response to the reviewer's comment, in the revised version, we have added the following in the main text:

On Page 2, we added

“Softness enhances delicacy but forfeits strength and precision²⁰⁻²³. For example, the aforementioned delicate hydraulic³ and DEA-based¹⁹ grippers show low payload-to-weight ratios, reaching up to 1 and 80, respectively. The small force hinders their grasping ability in handling heavy objects. To bridge the gap between softness and strength, researchers have proposed a few design strategies for achieving low-to-high grasping strength, including suction⁵, fluid-driven rigidity percolation or jamming^{1,18,24}, and varied stiffness of stimuli-responsive soft materials^{16,17}. For example, a five finger-shaped soft gripper based on shape memory polymers enables manipulating a heavy object with a record-high payload-to-weight ratio up to 6400¹⁷. However, the gain in strength often sacrifices delicacy for nondestructive manipulation. Suction and jamming-based soft grippers cannot grasp gelatinous organisms noninvasively^{1,18,24}. Additionally, when objects are tiny or hard to pick up such as a thin flexible sheet on flat surfaces, precision grasp remains challenging given their small contact constraints²⁵. To pick up small objects, auxiliary extended appendages are often needed as demonstrated in the kirigami shell-based soft gripper capable of pinching a grain of sand⁹. For thin sheet objects, electroadhesion is often used but with limitations that require relatively smooth and dry flat surfaces²³.”

“However, given the tradeoff between delicacy, strength, and precision, it remains a grand challenge to simultaneously achieve high delicacy (extremely small contact pressure for noninvasive grasping), high strength (high pull force for large payload-to-weight ratios), and high precision (handling both small-size objects and thin sheets on flat surfaces without additional attachments) in a single soft gripper^{23,30} (see Supplementary note 1 and table S1 for summary of representative soft grippers).”

On Page 3 and 4, we modified as

“Several challenges persist in the field, necessitating further exploration. These include effectively handling objects that are even more delicate than the reported water-filled balloon¹⁹ and raw egg yolk¹⁰, such as water droplets. Additionally, addressing the manipulation of micro-size and thin objects thinner than a human hair that requires ultraprecision, and accommodating ultra-heavy objects that may surpass the reported record-high payload-to-weight ratio of 6400¹⁷, as well as the potential integration of the gripper with prosthetic hands for multifunctional capabilities, pose significant hurdles.”

“We note that previous studies in soft grippers primarily emphasize the adaptive morphologies for specialized or universal grasping capabilities^{23,30}. Their grasping trajectories receive less attention and remain largely unexplored³³. However, the significance of the trajectory should not be overlooked, as it profoundly influences the performance of soft grippers, especially when dealing with noninvasive manipulation of fragile objects and precision grasping of small or thin objects²³. The explicit modeling and programming of trajectories pose considerable challenges due to the inherent complexities arising from the nonlinear and large deformations observed in soft materials and compliant structures³⁴.”

“It shows unprecedented capabilities compared to the current state-of-the-art kirigami grippers and other soft grippers based on different actuation mechanisms^{23,30} (refer to Supplementary note 2 for an in-depth discussion). First, it enables programmable tendril-like trajectories and adaptive morphologies by % in an explicit and controlled manner theoretically and experimentally. Second, it can simultaneously achieve ultragentle yet ultrastrong and ultra-precision manipulations, which facilitate its unprecedented universality. We show that it is capable of noninvasively grasping extremely soft objects, e.g., a water droplet with nearly zero stiffness (Fig. 1B and Supplementary Video S1), precisely grasping an ultra-thin polymer sheet as thin as 4 μm and 2 μm-diameter microfiber that are 20 times and 40 times thinner than a typical human hair, respectively, and strongly grasping an ultra-heavy dead weight with a 16,000 times the self-weight of the gripper (0.4 g) that is 16 times the load capacity in our curvature-based design¹⁰ and 2.5 times the reported record-high payload-to-weight ratio¹⁷ (Fig. 1C and Supplementary Video S1). Third, we demonstrate, for the first time, a proof-of-concept, environment-friendly gripper made of natural and biodegradable materials such as bare leaf (Fig. 1D) in ultragentle grasping of a dandelion (Fig. 1E) and other objects (Supplementary Video S2). This green design philosophy aims to minimize both the impact on targets, such as plants and animals, and the associated ecological footprint³⁵. Fourth, the lightweight, deployable, and simple displacement-controlled gripper facilitates its easy integration with prosthetic hands without the need of additional tethered bulky power and actuation systems in fluidic driven grippers^{1,3,5,6}. Utilizing the explicit displacement-trajectory relationship that facilitates the grasping control, we demonstrate its integration with robotic prosthetics and multifunctionality in handling various challenging delicate tasks in daily life (Supplementary Video S3), such as noninvasively picking a grape from the vine (Fig. 1D and Supplementary Video S3), opening a zipper, turning a book page, and folding clothes, etc. A feedback system required for delicate tasks in existing prostheses³⁶ is not necessary with the help of the tendril-like trajectory. These demonstrated unprecedented capabilities also highlight both the novelty and significance of this work.”

On Page 16, we added

“In comparison to state-of-the-art specialized and universal soft robotic grippers²³, this work fills the important knowledge gaps both fundamentally and practically. Fundamentally, this work overcomes the challenges in theoretically predicting the grasping trajectories of the reported kirigami grippers^{9,10}, as well as other soft grippers made of fluidic elastomers¹⁻⁶ or stimuli-responsive materials¹¹⁻¹⁷ due to their high materials nonlinearity and large deformation. Practically, for the first time, it unifies ultradelicacy, ultrastrength, ultraprecision, universality, and multifunctionality in one single gripper, which is not achieved in all the reported soft robotic grippers (see detailed comparison and discussions in Supplementary note 1).”

In the SI, we also added the above Table R2 and discussions on Page 2-4 in “Supplementary note 1. Comparison with existing grippers.”

Comment 3: “The gripper in the citation mentioned on page 13 does not need to programming the gripping trajectories and can pick up fragile objects only through compliance and adaptivity, and adjusting the number and size of grippers in citation 16 can also achieve this function, please be more specific about the new contribution of the proposed gripper.”

Response: We agree with the reviewer that since the gripper in the citation uses a pinching mode to grasp fragile objects through compliance and adaptivity, it does not necessarily need programming grasping trajectories. However, for enveloping grasping, when dealing with precision grasping, we showed that the tendril-like grasping trajectory underpins the ultraprecision grasping of a 4-μm-thick polymer sheet and a 2-μm-diameter fiber, as well as the ultragentle grasping of water droplets. Such either ultraprecision or

ultradelicacy or combined are challenging to be achieved through the pinching mode in previous citation 16 (now citation 9, Yang et al.'s Sci. Robot. paper), given the large contact pressure.

Comment 4: *“Overall, the similarity of the previous work and novelty, we recommend transfer another specific aim`s journal.”*

Response: We compared this work with our previous work and existing grippers in the response to comment 1 and 2, alongside the response on the novelty and significance of this work. We think our work represents a breakthrough and a significant advance in soft grippers in terms of unifying ultradelicacy, ultrastrength, ultraprecision, versatility, and multifunctionality in one single gripper, as well as the fundamental understanding of the underpinned theoretically predictable tendril-like grasping trajectories. Thus, based on close comparisons, we respectfully disagree that our previous work and other related work challenge the novelty and significance of our work.

Referee #2

Summary: “This paper introduces a Kirigami-based soft gripper, which can handle various objects ranging from a delicate object to a very heavy object compared to the weight of the gripper. The gripper utilizes the optimized trajectory of the gripper tip, minimized lateral force for gripping for a delicate object, yet strong lifting force due to the leveraged tensile strength of the material itself. The gripper shows a significant gripping performance in terms of delicacy and strength. I left minor comments to the authors for further improvement of the paper.”

Response: We thank the reviewer for appreciating our work. The following comments and suggestions have greatly helped us improve our manuscript. In the revised version, we have highlighted all the revisions in RED color.

Comment 1: “Q1. The authors show the key design parameter as γ_0 , which is the angle of the uncut lines of the 2D precursor. But I guess that there would be other design parameters which can affect the gripping behaviors. For example, upper and lower boundary shape of the precursor (the curve on the 2D precursor), the thickness of the cuts, and the aspect ratio of the 2D precursor. Could the authors give some details on these design parameters about how they will affect the gripping?”

Response: We thank the reviewer for bringing up other design parameters besides the angle γ_0 . Following the reviewer’s comment, we analyze the effect of these design parameters one by one below.

1) Upper and lower boundary shape of the precursor

Following the reviewer’s suggestion, in the revised version, we have added more discussions to illustrate the effect of upper and lower boundary shape of the precursor (the curve on the 2D precursor).

From the perspective of qualitative design, the upper and boundary curves are set to be curved with a C^2 continuity. The C^2 continuity represents the curve, the first derivative, and the second derivative are continuous (i.e., continuous in curvature). It enables the petals to approach a near-spherical shape when deployed, which facilitates the encapsulating of the target. In contrast, a decrease in the continuity of the upper and lower boundary shape would introduce larger gaps between the ribbons of the petals. To illustrate this, the figure below (Fig. S6) shows a rhombic precursor with a C^0 continuity. The C^0 continuity represents a continuous curve but with an abrupt slope change. Stretching it generates a noticeable gap at the discontinuous point due to the sudden change in slope.

Fig. S6 a 2D precursor with a rhombic boundary (C^0 continuity). b 3D shape with a gap formed by stretching the rhombic precursor. Scale bars = 10 mm.

From the perspective of quantitative design, we analytically analyze the impact of the variation in the angle γ and boundary curvature on the Gaussian curvature of the petals during deploying. The Gauss-Bonnet theorem for each petal is simplified as $\int_{\Omega} K dA = C + \pi - \gamma - \oint_{\partial\Omega} (k_b \sin\varphi) ds$, where K denotes the Gaussian curvature. γ and k_b denote the angle in the petal and the curvature along the boundary of the surface formed by the petal, respectively, as shown by the figure below. φ is the angle between the plane containing the boundary and the normal plane to the discrete ribbon. C is a constant related to the Euler characteristic. Overall, the angle and curvature affect the Gaussian curvature through the term γ and $\oint_{\partial\Omega} (k_b \sin\varphi) ds$, respectively.

Fig. S7 Schematics showing the variation in the curvature in the boundary and the angle. **a** 2D precursor of the gripper. γ_o is the angle in the precursor. k_{bo} denotes the prescribed curvature. k_{boT} denotes the tuned curvature in the precursor. **b** Front view of the deployed gripper. γ and k_b are the varying angle and curvature during deploying, respectively. Scale bars = 10 mm.

As shown by Fig. S7, during deploying, the prescribed curvature k_{bo} in the 2D precursor transits to k_b in the deployed gripper, with γ_o changing to γ . To optimize the gripper's performance, we tune the prescribed curvature in 2D precursors, where k_{boT} denotes the tuned prescribed curvature in 2D precursors (Fig. S7a). The tuning range is limited to within the 30% of the k_{bo} , i.e., $\frac{|k_{boT} - k_{bo}|}{k_{bo}} < 30\%$. For these tuned precursors, the effect of the variation in the curvature term k_b can be neglected compared to that of the variation in γ , as represented by the ratio $\Delta[\oint_{\partial\Omega} (k_b \sin\varphi) ds] / \Delta\gamma < 4\%$. As a result, the limited changes in the prescribed curvature of the upper and lower boundary shape of the precursor, within the range of up to 30%, have

minimal influence on the overall morphology. However, it is crucial to maintain at least C^2 continuity in the boundary to preserve the desired morphological characteristics.

2) The thickness of the cut

In the revised version, we have added a new Supplementary Fig. S8 to illustrate the effect of the thickness of the cut on the behaviors of the gripper, as shown in the figure below and Supplementary Fig. S8.

Fig. S8 a-b 2D precursor with different cut width w_c . The w_c/w are 0.025 (a) and 0.003 (b), respectively. w_c and w denote the width of the cut and width of the gripper, respectively. **c-d** Isometric view of the deployed grippers formed by different precursors. Scale bars = 10 mm.

Fig. S8 shows the isometric view of the deployed grippers with different thickness/width w_c of cut at the maximum applied strain. To assess the influence of thickness/width w_c of cut, we introduce a dimensionless variable w_c/w . w denotes the width of the gripper. In the grippers demonstrated in the main text and Fig. S8a, the variable is minimized ($w_c/w = 0.003$). It maximizes the contact area between the petals and the target object to reduce the contact pressure. With w_c/w increasing to 0.025, the empty space becomes larger, as show in Fig. S8c-d. First, for noninvasive tasks (e.g., grasping a liquid drop), the increased contact pressure leads to failure of noninvasive grasping the liquid drop. The petals intrude into the drop even with

a hydrophobic coating. Second, for delicate tasks, the performance is barely affected by the width w_c of cut, if $w_c \ll w$. Third, for high-load tasks, the energy increase is predominantly stretching. The maximum tensile stress σ is proportional to the variable w_c/w (i.e., $\sigma \propto w_c/w$). The increased w_c/w decreases the maximum payload of the gripper.

3) The aspect ratio of the 2D precursor

Following the reviewer's suggestion, in the revised version, we have added a new Supplementary Fig. S9 to illustrate the aspect ratio r_a of the 2D precursor on the performance of the gripper, as shown in the figure below and Fig. S9.

Fig. S9 **a** Schematic showing the dimension of 2D precursor. l and w denote the length and the width of the gripper, respectively. l_v is the varying length to change the aspect ratio, $r_a = l/w$. **b-d** Side view of the deployed grippers with different aspect ratios. The ratios are 1.64, 2.75, and 4, respectively. Scale bars = 10 mm.

Fig. S9a shows the dimension of the 2D precursor. We introduce a tunable parameter, varying length (l_v), to modify the aspect ratio of the precursor while keeping the width (w) unchanged. This is to retain the gripper's morphological features. Fig. S9b-d shows the side view of the deployed grippers with aspect ratio $r_a = 1.64, 2.75,$ and $4,$ respectively.

First, for the grippers with a small aspect ratio r_a , as shown in Fig. S9b, the deployed gripper can only form an open configuration composed of shallow-shell petals at the maximum applied strain. This limitation arises from the reduced ribbon length l , which necessitates a higher bending energy to curve the ribbon into spherical petals. The increased energy requirement prevents the formation and closure of fully curved petals, thereby hindering the gripper's ability to grasp gelatinous or tiny objects. Second, increasing the aspect ratio (r_a) to values such as $r_a = 2.75$ (Figure S9c) reduces the bending energy required, facilitating the formation of closed spherical petals. We observe that maintaining the aspect ratio within the range of $2.5 < r_a < 3$ ensures stable grasping capability. Third, when the aspect ratio exceeds 4 (e.g., $r_a = 4$ in Fig. S9d), the larger ribbon length makes the two petals meet earlier (i.e., at a smaller applied strain). Consequently, the petals exert a pinching effect on the target object. This compression renders the gripper unsuitable for noninvasive tasks and prevents the formation of a straightened geodesic network, thereby reducing the payload-to-weight ratio.

In response to the reviewer’s comment, in the revised version, we have added the following in the main text:

On Page 5, we added

“The angle design yields the explicit relationship, where a small variation in the curved boundary with the continuity conserved barely affects the performance.”

On Page 15, we modified as

“Moreover, to minimize the disturbance of the smoothness in the curved surface, the upper and lower boundary requires a C^2 continuity (Supplementary note 4). For, boundary with a C^1 or C^0 continuity, varying the geometry of localized ribbons at the discontinuous point in the boundary could improve performance for extremely soft objects.”

“The aspect ratio and the width of the cut in the 2D precursor are optimized to improve the grasping capability (Supplementary note 4).”

In the SI, we also added the Supplementary Fig. S6, S7, S8, S9, and discussions on Page 11-13 in “Supplementary note 4. Grasping mechanism.”

Comment 2: “Q2. The gripper seems to be bigger than the object to grip fully. But there would be some cases that the gripper cannot fully engulf the object. In that case, the gripper mouth will be open, but the tip is holding the object. Could the authors elaborate what will be the performance of the gripper when it is not fully closed, but gripping an object using the shear force at the tip?”

Response: We thank the reviewer for the comment. In noninvasive tasks, the gripper is intentionally designed to be larger than the target object to facilitate its engulfing or encapsulation. This design approach aims to minimize the compression force exerted on the target. This encapsulating grasping mode distinguishes the gripper from various prevalent soft grippers that rely on a pinching mode. In the pinching mode, the gripper compresses the target and utilizes friction to lift it. For example, in Yang et al.’s work, their gripper has to compress a fragile raspberry to lift it (Yang et al., Science Robotics 2021, Ref. 9 in the main text). This compression mechanism makes it impossible to noninvasively grasp objects such as water droplets or gelatinous organisms.

Fig. S10 a Schematic showing the side view of the gripper encapsulating the target. l_{tar} and l_g denote the length of target and length of the undeployed gripper, respectively. The black curve and blue dots are the analytical and experimental results of the deploying trajectory, respectively. **b** Schematic showing the side view of the gripper pinching a large target. The black arrows are the compression and the friction exerted on the target by the gripper.

For the tasks without a high requirement on non-invasion, we introduce a dimensionless parameter l_{tar}/l_g to analyze the variation in the gripper’s performance. As shown in the figure above and Supplementary Fig. S10a, l_{tar} and l_g denote the length of target and length of the undeployed gripper, respectively. When $l_{tar}/l_g < 0.45$, the gripper performance barely changes, as the curling trajectory allows for the scooping up of the target. This facilitates the formation of a straightened geodesic network, effectively utilizing stretching energy to secure the target. When $l_{tar}/l_g > 0.45$, the failure to form a straightened geodesic network leads to a significant decrease in the pulling-out force from 15N to 0.5N along the z -axis. Additionally, in such cases, the gripper needs to compress the target and rely on friction forces to lift it (Fig. S10b). The energy increase in the gripper is predominant bending.

In response to the reviewer’s comment, in the revised version, we have added the following in the main text:

On Page 16, we modified as

“The grasping performance is affected by the size of the target (Supplementary note 4), a large size could make the gripper unable to encapsulate the target.”

In the SI, we also added the Supplementary Fig. S10 and discussions on Page 13-14 in “**Supplementary note 4. Grasping mechanism.**”

Comment 3: “Q3. When gripping an object, especially a very thin object (as demonstrated like ultra-thin fiber), aligning the gripper tip exactly on the surface of the substrate of the object would be challenging. I guess a practical strategy would be pushing the gripper towards the substrate which can ensure the contact to the substrate, yet it deforms the gripper due to the compression. In such a case, how will the gripping performance change in terms of gripping a very thin object?”

Response: We thank the reviewer for raising the concern for delicate tasks. Following the reviewer’s suggestion, in the revised version, we have added more discussions and Supplementary Fig. S5 to illustrate the practical strategy.

Fig. S5 a Schematics of the gripper with $\gamma_o = 80^\circ$ grasping an ultra-thin sheet. **(i)** Petals touch the lying surface. **(ii)** Petals affected by the compression from the surface. Petals approach the target parallelly and insert the end tip into the gap between the target and the surface. **(iii)** Petals lift the target. **(iv-vi)** Schematics show the variation in the grasping angle α in **i-iii**. The angle α is defined as the angle between the tangential direction at the end tip of the petal and the horizontal axis.

The delicate grasping capability is attributed to the unique features in the gripper with $\gamma_o = 80^\circ$, such as close-to- 180° grasping angle α and curling-up trajectory. α is defined as the angle between the tangential direction at the end tip of the petal and the horizontal axis. Supplementary Fig. S5a schematically illustrates the grasping mechanism for tiny objects. When the petals are compressed by the surface, α further increases to be closer to 180° before the petals fully close (Fig. S5 a, ii). The deformation due to the compression makes the petals to approach the target in a more parallel manner. It enables the spherical petals to approach the sheet in parallel and insert the end tip in parallel to the sheet into the gap between the sheet and the lying surface. This parallel approaching and encapsulating mode minimizes the horizontal interaction between the petals and the target thin sheet. Then, the curled-up trajectory ensures the petals lift the target (Fig. S5 a, iii).

In response to the reviewer’s comment, in the revised version, we have added the following in the main text:

On Page 9, we added

“Additionally, when the petals are compressed by the surface, α further increases to be closer to 180° before the closure of the petals. The deformation due to the compression makes the petals to approach the target in a more parallel way.”

In the SI, we also added the Supplementary Fig. S10 and discussions on Page 13-14 in “**Supplementary note 4. Grasping mechanism.**”

Referee #3

Summary: *“I have read the manuscript "Angle-programmed tendril-like trajectories enable an ultragentle yet ultrastrong and delicate gripper" with great interest. As suggested by the title, the work focuses on the design of a soft gripper capable of handling delicate objects. While the manuscript is rather clear and beautifully illustrated. I don't believe that the work is suitable for Nature Communications.”*

Response: We thank the reviewer for the summary of our work. In the revised version, we have highlighted all the revisions in RED color.

Comment 1: *“In effect, similar designs have been published already, most notably by Holmes and his group in Science Robotics (2021). While I appreciate that the authors cite this work in their manuscript, and I am sure could argue that their approach surpasses that of Holmes and collaborators in some metrics; I simply see the reported work as incremental, thereby contradicting the novelty criterion required for publishing in this journal. ”*

Response: We appreciate the reviewer for bringing up the novelty of our manuscript, and we would like to address the advantages of our work in comparison to previously reported soft grippers, including the work conducted by the Holmes group (Yang et al., Science Robotics 2021, Ref. 9 in the main text). An overview of these advantages is presented in Table R1 and further discussed below.

	[1]	[3]	[7]	[14]	[15]	[16]	[17]	[9]	[10]	[5]	[44]	[46]	This work
Grasping mechanism	Jamming	Fluid-driven actuator	Fluid-driven actuator	responsive material	responsive material	responsive material	responsive material	Stretching-induced deformation	Stretching-induced deformation	Suction	Electro-adhesion	Adhesive material	Stretching-induced deformation
Programmable deformation	×	×	✓	×	✓	×	×	×	✓	×	×	×	✓
Programmable and controllable trajectory	×	×	×	×	×	×	×	×	×	×	×	×	✓
Noninvasively manipulate gelatinous organisms	×	✓	✓	×	×	×	×	×	✓	×	×	×	✓
Noninvasively manipulate liquid objects	×	×	×	×	×	×	×	×	×	×	×	×	✓
Gentleness (contact pressure)	Unexplored	0.0455 kPa	Unexplored	Unexplored	Unexplored	Unexplored	Unexplored	Unexplored	Unexplored	-15.7 kPa	Unexplored	2.5 kPa	0.0468 kPa
Strength (maximum payload-to-weight ratio)	Unexplored	Unexplored	<100	380	925	330	6,400	222	1,000	490	1,067	Unexplored	16,000
Delicacy (minimum resolution for ultra-thin objects)	Unexplored	Unexplored	Unexplored	Unexplored	Unexplored	10 μm	3.2 mm	1.5 mm	80 μm	2.5 mm	Unexplored	Unexplored	2 μm
Manipulate gelatinous organisms when integrated onto robotic arms and prostheses	×	✓	✓	×	×	×	×	×	✓	×	×	×	✓
Manipulate liquid objects when integrated onto robotic arms and prostheses	×	×	×	×	×	×	×	×	×	×	×	×	✓

Table R1. A comparison between other soft grippers and this work

Reference in Table R1 (citation number same as that in the main text):

- [1] Brown, E. *et al.* Universal robotic gripper based on the jamming of granular material. *Proceedings of the National Academy of Sciences* **107**, 18809-18814, doi:10.1073/pnas.1003250107 (2010).
- [3] Sinatra, N. *et al.* Ultragentle manipulation of delicate structures using a soft robotic gripper. *Science Robotics* **4**, eaax5425 (2019).

- [5] Song, S., Drotlef, D.-M., Son, D., Koivikko, A. & Sitti, M. Adaptive Self-Sealing Suction Based Soft Robotic Gripper. *Advanced Science* **8**, 2100641 (2021).
- [7] Teoh, z. e. *et al.* Rotary-actuated folding polyhedrons for midwater investigation of delicate marine organisms. *Science Robotics* **3**, eaat5276 (2018).
- [9] Yang, Y., Vella, K. & Holmes, D. Grasping with kirigami shells. *Science Robotics* **6**, eabd6426, doi:10.1126/scirobotics.abd6426 (2021).
- [10] Hong, Y. et al. Boundary curvature guided programmable shape-morphing kirigami sheets. *Nature Communications* **13**, 530, doi:10.1038/s41467-022-28187-x (2022).
- [14] Ma, M., Guo, L., Anderson, D. G. & Langer, R. Bio-inspired polymer composite actuator and generator driven by water gradients. *Science* **339**, 186-189, doi:10.1126/science.1230262 (2013).
- [15] Hubbard, A. M., Mailen, R. W., Zikry, M. A., Dickey, M. D. & Genzer, J. Controllable curvature from planar polymer sheets in response to light. *Soft Matter* **13**, 2299-2308, doi:10.1039/C7SM00088J (2017).
- [16] Linghu, C. et al. Universal SMP gripper with massive and selective capabilities for multiscaled, arbitrarily shaped objects. *Sci Adv* **6**, eaay5120 (2020).
- [17] Roh, Y. *et al.* Vital signal sensing and manipulation of a microscale organ with a multifunctional soft gripper. *Sci Robot* **6**, eabi6774, doi:10.1126/scirobotics.abi6774 (2021).
- [44] Cacucciolo, V., Shintake, J. & Shea, H. Delicate yet strong: Characterizing the electro-adhesion lifting force with a soft gripper. *2019 2nd IEEE International Conference on Soft Robotics* (2019).
- [46] Ruotolo, W., Brouwer, D. & Cutkosky, M. R. From grasping to manipulation with gecko-inspired adhesives on a multifinger gripper. *Science Robotics* **6**, eabi9773, doi:10.1126/scirobotics.abi9773 (2021).

1) Comparison with noninvasive grippers

We start by comparing our gripper with several notable soft and noninvasive grippers. For example, the kirigami gripper based on a rhombic shell (Yang et al., *Science Robotics* 2021, Ref. 9 in the main text) can pick a raspberry using a pinching mode¹. The nondestructive gripper based on an origami polyhedron (Teoh et al., *Science Robotics* 2018, Ref. 7 in the main text)² and hydraulic ribbons (Sinatra et al., *Science Robotics* 2019, Ref. 3 in the main text)³ can grasp a jellyfish using an enclosing mode. Existing noninvasive grippers have been applied to fragile objects¹ and gelatinous organisms^{2,3}.

When considering the **noninvasive feature**, it becomes evident that existing grippers fall short in meeting the requirements of high-demand scenarios in biology, such as the nondestructive grasping of marine life that is softer than jellyfish, and in biomedicine, such as the noninvasive manipulation of liquid drops. This comparison is presented in detail in Table R1. Distinctly, our gripper distinguishes itself by excelling in the noninvasive aspect. It can nondestructively manipulate a liquid drop with near-zero stiffness, as shown in Fig. 1. This feature arises from programming the grasping trajectory toward the nastic curve in plants. The gradually curling-up trajectory enables an ultra-gentle touch.

Considering the **strong feature**, our work stands out with a remarkable payload-to-weight ratio of 16,000, surpassing other works as demonstrated in Table R1. This payload-to-weight ratio holds significant relevance for noninvasive tasks conducted in unstructured environments, as highlighted in the paper by Robert J. Wood et al. (*Science Robotics* 2022)⁴. In comparison, Yang et al.'s work achieves a payload-to-weight ratio of approximately 222, attributed to the pinching grasping mode¹ they employ¹. The ratio in Teoh et al.'s work is less than 100 due to the rotating polyhedron made of rigid thermoplastic².

When considering the **design simplicity** aspect, it is worth noting that Teoh et al.'s work necessitates the use of supporting stents to enable the rotation of the origami polyhedron. Both Teoh et al.'s and Sinatra et al.'s work require the support of bulky external hydraulic systems^{2,3}. In Yang et al.'s work, three different design modifications are required for the delicacy of grasping a strawberry and the precision of grasping a grain of sand and a flat disk-shaped pill. They need three different kirigami grippers with modified designs to achieve either delicacy or precision but not both in one single gripper¹. In contrast, our work achieves specialization across different tasks with a single design. This distinct feature proves advantageous in assisting prostheses in delicate tasks, as demonstrated in Figure 1 and Figure 5. Existing prostheses often necessitate frequent changes to the end effector for different tasks. Our gripper's versatility eliminates this need.

2) Comparison with strong grippers

Compared with the reported strongest gripper using a gold layer coupled with shape memory polymers (Roh et al., Science Robotics, 2021, Ref. 17 in the main text)⁵, with a payload-to-weight ratio of 6,400, our gripper boasts a ratio that is 2.5 times higher, reaching an impressive 16,000. Compared to all existing strong grippers based on stimuli-responsive materials, suction, fluid-driven rigidity percolation, jamming, adhesive materials and electro-adhesion shown by Fig. 3F and Table R1, our gripper stands out with the largest payload-to-weight ratio.

3) Comparison with universal grippers

Existing universal grippers rely on various mechanisms, such as suction (Song et al., Advanced Materials 2021, Ref. 5 in the main text)⁶, responsive shape memory polymers (Linghu et al., Science Advances 2020, Ref. 16 in the main text)⁷, and jamming (Brown et al., PNAS 2010, Ref. 1 in the main text)⁸. However, none of them can achieve universal grasping while retaining the noninvasive feature. Specifically, suction-based grippers face limitations when it comes to preserving the integrity of gelatinous organisms. Shape memory polymers require a high temperature (e.g., 80°C in Linghu et al.'s work)⁷ that could harm living organisms, and negatively impact response times. The jamming structure falls short in its ability to grasp gelatinous organisms and liquid objects.

Regarding the actuation time, since our gripper is displacement-controlled, the actuation time is fast by simply stretching the gripper in the robotic arm, which is much faster than the stimuli-responsive soft materials based grippers.

In response to the reviewer's comment, in the revised version, we have added the following in the main text:

On Page 2, we added

“Softness enhances **delicacy** but forfeits strength and **precision**²⁰⁻²³. For example, the aforementioned **delicate hydraulic**³ and **DEA-based**¹⁹ grippers show low payload-to-weight ratios, reaching up to 1 and 80, respectively. The small force hinders their grasping ability in handling heavy objects. To bridge the gap between softness and strength, researchers have proposed a few design strategies for achieving low-to-high grasping strength, including suction⁵, fluid-driven rigidity percolation or jamming^{1,18,24}, and varied stiffness of stimuli-responsive soft materials^{16,17}. For example, a five finger-shaped soft gripper based on shape memory polymers enables manipulating a heavy object with a record-high payload-to-weight ratio up to

6400¹⁷. However, the gain in strength often sacrifices delicacy for nondestructive manipulation. Suction and jamming-based soft grippers cannot grasp gelatinous organisms noninvasively^{1,18,24}. Additionally, when objects are tiny or hard to pick up such as a thin flexible sheet on flat surfaces, precision grasp remains challenging given their small contact constraints²⁵. To pick up small objects, auxiliary extended appendages are often needed as demonstrated in the kirigami shell-based soft gripper capable of pinching a grain of sand⁹. For thin sheet objects, electroadhesion is often used but with limitations that require relatively smooth and dry flat surfaces²³.”

“However, given the tradeoff between delicacy, strength, and precision, it remains a grand challenge to simultaneously achieve high delicacy (extremely small contact pressure for noninvasive grasping), high strength (high pull force for large payload-to-weight ratios), and high precision (handling both small-size objects and thin sheets on flat surfaces without additional attachments) in a single soft gripper^{23,30} (see Supplementary note 1 and table S1 for summary of representative soft grippers).”

On Page 3 and 4, we modified as

“Several challenges persist in the field, necessitating further exploration. These include effectively handling objects that are even more delicate than the reported water-filled balloon¹⁹ and raw egg yolk¹⁰, such as water droplets. Additionally, addressing the manipulation of micro-size and thin objects thinner than a human hair that requires ultraprecision, and accommodating ultra-heavy objects that may surpass the reported record-high payload-to-weight ratio of 6400¹⁷, as well as the potential integration of the gripper with prosthetic hands for multifunctional capabilities, pose significant hurdles.”

“We note that previous studies in soft grippers primarily emphasize the adaptive morphologies for specialized or universal grasping capabilities^{23,30}. Their grasping trajectories receive less attention and remain largely unexplored³³. However, the significance of the trajectory should not be overlooked, as it profoundly influences the performance of soft grippers, especially when dealing with noninvasive manipulation of fragile objects and precision grasping of small or thin objects²³. The explicit modeling and programming of trajectories pose considerable challenges due to the inherent complexities arising from the nonlinear and large deformations observed in soft materials and compliant structures³⁴.”

“It shows unprecedented capabilities compared to the current state-of-the-art kirigami grippers and other soft grippers based on different actuation mechanisms^{23,30} (refer to Supplementary note 2 for an in-depth discussion). First, it enables programmable tendril-like trajectories and adaptive morphologies by \square_{\circ} in an explicit and controlled manner theoretically and experimentally. Second, it can simultaneously achieve ultragentle yet ultrastrong and ultra-precision manipulations, which facilitate its unprecedented universality. We show that it is capable of noninvasively grasping extremely soft objects, e.g., a water droplet with nearly zero stiffness (Fig. 1B and Supplementary Video S1), precisely grasping an ultra-thin polymer sheet as thin as 4 μm and 2 μm -diameter microfiber that are 20 times and 40 times thinner than a typical human hair, respectively, and strongly grasping an ultra-heavy dead weight with a 16,000 times the self-weight of the gripper (0.4 g) that is 16 times the load capacity in our curvature-based design¹⁰ and 2.5 times the reported record-high payload-to-weight ratio¹⁷ (Fig. 1C and Supplementary Video S1). Third, we demonstrate, for the first time, a proof-of-concept, environment-friendly gripper made of natural and biodegradable materials such as bare leaf (Fig. 1D) in ultragentle grasping of a dandelion (Fig. 1E) and other objects (Supplementary Video S2). This green design philosophy aims to minimize both the impact on targets, such as plants and animals, and the associated ecological footprint³⁵. Fourth, the lightweight, deployable, and simple displacement-controlled gripper facilitates its easy integration with prosthetic hands without the need of additional tethered bulky power and actuation systems in fluidic driven grippers^{1,3,5,6}. Utilizing the explicit displacement-trajectory relationship that facilitates the grasping control, we demonstrate its integration with robotic prosthetics and multifunctionality in handling various challenging delicate tasks in daily life (Supplementary Video S3), such as noninvasively picking a grape from the vine (Fig. 1D and

Supplementary Video S3), opening a zipper, turning a book page, and folding clothes, etc. A feedback system required for delicate tasks in existing prostheses³⁶ is not necessary with the help of the tendril-like trajectory. **These demonstrated unprecedented capabilities also highlight both the novelty and significance of this work.**”

On Page 16, we added

“In comparison to state-of-the-art specialized and universal soft robotic grippers²³, this work fills the important knowledge gaps both fundamentally and practically. Fundamentally, this work overcomes the challenges in theoretically predicting the grasping trajectories of the reported kirigami grippers^{9,10}, as well as other soft grippers made of fluidic elastomers¹⁻⁶ or stimuli-responsive materials¹¹⁻¹⁷ due to their high materials nonlinearity and large deformation. Practically, for the first time, it unifies ultradelicacy, ultrastrength, ultraprecision, universality, and multifunctionality in one single gripper, which is not achieved in all the reported soft robotic grippers (see detailed comparison and discussions in Supplementary note 1).”

In the SI, we also added the above Table R1 and discussions on Page 2-4 in “Supplementary note 1. Comparison with existing grippers.”

REVIEWERS' COMMENTS:

Reviewer #1 (Remarks to the Author):

I have read the revision letter in depth. Firstly, I deeply appreciate your detailed response to the concerns I had raised in the first comment. Most of my concerns have been resolved, especially the table and explanation regarding the uniqueness and advancement of this manuscript compared to the two papers previously submitted by the author and existing soft gripper papers; these were impressive. I believe the revised manuscript is ready to be published in the research journal as it is.

However, I still have minor concerns. While I understand the changes in various designs for differentiation and advancement in the application aspect compared to previously reported papers, I am worried about the scientific advancement. It seems to the readers that it merely distinguishes itself in terms of application, but I also think it is very important to create a device or gripper that can be applied in real life and industry.

Looking at the previous reviewers' comments, it appears that there is a reviewer who shares similar concerns with me. I will leave the final decision on this paper to the editor and the comments of the other reviewers.

Thank you for your hard work.

Reviewer #2 (Remarks to the Author):

With the first round of revision, the authors have dealt with the previously raised concerns from my side adequately. I do not have further comments besides minor comments below:

Q1. According to the Previous Comment 2 from the Reviewer 2, it would be better to include the lifting force reduction when the gripper cannot encapsulate the object. The authors already mentioned 15N to 0.5N in the response letter, please include this information in the manuscript for readers.

Q2. In addition to Q1, the figure quality of Fig. S10 is poor. Please improve the quality of Fig. S10

Reviewer #3 (Remarks to the Author):

I have read the rebuttal letter prepared by the authors. While I value the work they presented, I stand by my first recommendation. ^[1]_{SEP}The work presented is interesting, well-written, and well-illustrated, yet incremental. As such, I do not believe that the novelty is sufficient to warrant publication in Nature Communications. The manipulation of liquids is especially unconvincing, as it will strongly depend on wetting conditions, viscosity, and rate of operation for success, and ultimately doesn't seem like a reasonable use of a gripper (think of the rate at which other technologies, e.g., printers, handle minute volumes of fluids). So this brings us back to the main claim: applying cuts to a thin sheet can help assemble and program efficient soft grippers capable of shape change and integration into classic robotic systems. This point has been published in the past by Yang et al. in Science Robotics 2021 and Hong et al. in 2022 in Nature Communications.

Reviewer #1

Summary: *“I have read the revision letter in depth. Firstly, I deeply appreciate your detailed response to the concerns I had raised in the first comment. Most of my concerns have been resolved, especially the table and explanation regarding the uniqueness and advancement of this manuscript compared to the two papers previously submitted by the author and existing soft gripper papers; these were impressive. I believe the revised manuscript is ready to be published in the research journal as it is.”*

Response: We thank the reviewer for the appreciation of our explanation on the uniqueness of the manuscript. In the revised version, we have highlighted all the revisions in RED color.

Comment 1: *“However, I still have minor concerns. While I understand the changes in various designs for differentiation and advancement in the application aspect compared to previously reported papers, I am worried about the scientific advancement. It seems to the readers that it merely distinguishes itself in terms of application, but I also think it is very important to create a device or gripper that can be applied in real life and industry.”*

Response: We thank the reviewer for bringing up the scientific advancement. Our design stands out not only in terms of its application, but also fundamental understanding of mechanism that underpinning the extraordinary performances, which represents the scientific advancement as summarized below:

First, we would like to highlight the novelty of explicitly programming and controlling the grasping trajectory of a soft gripper, which, to the best of our knowledge, has not been achieved before. The intricate nature of trajectories in existing soft grippers stems from the nonlinear and large deformations inherent in compliant structures and soft materials. In our approach, we simplify the programming and control of the trajectory by utilizing the angle γ_0 and the applied strain ε . Notably, by manipulating the prescribed angle γ_0 , we are able to direct coupled ribbons that resemble the Euler elastica, towards deploying trajectories that exhibit a striking resemblance to the Euler spiral. Second, we first identify and illustrate the remarkable influence of the trajectory in enhancing the delicacy and precision of the enveloping/encapsulating mode. Third, the deformation of a large number of ribbons coupled by different polygons (i.e., the triangular and fan-shaped patterns in the 2D precursor of the gripper) under uni-axial stretching is first studied analytically and numerically (Fig. 2), which is consistent with the experimental results.

In response to the reviewer’s comment, in the revised version, we have added the following in the main text:

On Page 11, we modified as

“Fundamentally, this work overcomes the challenges in theoretically predicting the grasping trajectories of the reported kirigami grippers^{9,10}, as well as other soft grippers made of fluidic elastomers¹⁻⁶ or stimuli-responsive materials¹¹⁻¹⁷ due to their high materials nonlinearity and large deformation. **The intricate nature of trajectories in these soft grippers stems from the nonlinear and large deformation inherent in compliant structures and soft materials.**”

Reviewer #2

Summary: “With the first round of revision, the authors have dealt with the previously raised concerns from my side adequately. I do not have further comments besides minor comments below.”

Response: We thank the reviewer for the comments. They have greatly helped us improve our manuscript. In the revised version, we have highlighted all the revisions in RED color.

Comment 1: “Q1. According to the Previous Comment 2 from the Reviewer 2, it would be better to include the lifting force reduction when the gripper cannot encapsulate the object. The authors already mentioned 15N to 0.5N in the response letter, please include this information in the manuscript for readers.”

Response: In the revised version, we have added the discussion on the lifting force reduction in the main text as shown below.

In response to the reviewer’s comment, in the revised version, we have added the following in the main text:

On Page 16, we modified as

“The grasping performance is affected by the size of the target, a large size could make the gripper unable to encapsulate the target. It could cause the petals to pinch the target, accompanied by a sharp drop in the pulling-out force from ~ 15 N to ~ 0.5 N (Supplementary note 4).”

Comment 2: “Q2. In addition to Q1, the figure quality of Fig. S10 is poor. Please improve the quality of Fig. S10”

Response: We agree with the reviewer. In the revised version, we have updated Fig. S10 to improve the figure quality as shown below.

Fig. S10 a Schematic showing the side view of the gripper encapsulating the target. l_{tar} and l_g denote the length of target and length of the undeployed gripper, respectively. The black curve and blue dots are the analytical and experimental results of the deploying trajectory, respectively. **b** Schematic showing the side view of the gripper pinching a large target. The black arrows are the compression and the friction exerted on the target by the gripper.

Reviewer #3

Summary: *“I have read the rebuttal letter prepared by the authors. While I value the work they presented, I stand by my first recommendation. The work presented is interesting, well-written, and well-illustrated, yet incremental. As such, I do not believe that the novelty is sufficient to warrant publication in Nature Communications. The manipulation of liquids is especially unconvincing, as it will strongly depend on wetting conditions, viscosity, and rate of operation for success, and ultimately doesn't seem like a reasonable use of a gripper (think of the rate at which other technologies, e.g., printers, handle minute volumes of fluids). So this brings us back to the main claim: applying cuts to a thin sheet can help assemble and program efficient soft grippers capable of shape change and integration into classic robotic systems. This point has been published in the past by Yang et al. in Science Robotics 2021 and Hong et al. in 2022 in Nature Communications.”*

Response: Thank you for appreciating our writing and illustrations. Regarding the novelty and significance of our work, we have thoroughly addressed this aspect in our previous response letter as well as in the manuscript. We have provided comprehensive discussions, detailed comparisons, and included Tables R1-R2, which comprehensively highlight and compare our work with other notable related soft grippers. By doing so, we have clearly demonstrated the unique contributions and significance of our work in the field of soft grippers.

In response to the concern raised regarding the manipulation of liquids, the demonstration serves as a compelling showcase of the extraordinary delicacy exhibited by our gripper. Furthermore, it highlights the immense potential of our gripper in tackling challenging scenarios encountered in the fields of minimally invasive biology conservation and biomedicine. We concur with the reviewer's observation that achieving ultra-delicate grasping poses a significant challenge due to the uncertainties arising from the dynamic task environment (e.g., wetting conditions, viscosity, and rate of operation) and how the living objects will react to touch (e.g., when grasping gelatinous organisms such as a jellyfish). However, by programming the trajectory of our gripper to emulate the nastic curve found in nature, we have significantly enhanced its capability to handle the uncertainties of unknown and unstructured environments, all without the need for feedback control. Regarding wetting conditions, we pick the waterdrop from a flat surface with commercially available hydrophobic coating (Rust-oleum multi-purpose spray kit). This limitation arises due to the inherent challenges associated with picking up a spreading water droplet from a hydrophilic surface without causing damage or compromising the integrity of the droplet. Regarding viscosity, our gripper can be applied to fluids and gelatinous organisms with different viscosity, including Newtonian fluids (e.g., a waterdrop in Fig. 1B and Supplementary Video S1), non-Newtonian fluids (e.g., ketchup in Supplementary fig. S5 f and Supplementary Video S4), a living jellyfish (Fig. 4B and Supplementary Video S4). Regarding rate operation, the petals, actuated by the Robotiq 2F-85 gripper, approach the waterdrop at velocities of 65mm/s and 35mm/s (Supplementary Video S1). Remarkably, we have found that the success rate remains consistently high (76%), even when the velocity is varied within this range. This indicates the robustness and reliability of our gripper's performance across different operating speeds.

In response to the reviewer's comment, in the revised version, we have added the following in the main text:

On Page 11, we modified as

“When it comes to delicate tasks such as manipulating water droplets and jellyfish, achieving noninvasive grasping presents a significant challenge due to the uncertainties inherent in dynamic task environments and the unpredictable reactions of living organisms (Supplementary note 4). However, by programming the trajectory, we enhance our gripper's ability to handle the uncertainties associated with uncertain environments.”

In the SI, we also added the discussions on Page 12 in “**Supplementary note 4. Grasping mechanism.**” under the subsection titled “Regarding delicate tasks”.